# MAKE LLMS BETTER ZERO-SHOT REASONERS: STRUCTURE-ORIENTATED AUTONOMOUS REASONING

## ABSTRACT

Zero-shot reasoning methods with Large Language Models (LLMs) offer significant advantages including great generalization to novel tasks and reduced dependency on human-crafted examples. However, the current zero-shot methods still have limitations in complex tasks, e.g., answering questions that require multi-step reasoning. In this paper, we address this limitation by introducing a novel structure-oriented analysis method to help LLMs better understand the question and guide the problem-solving process of LLMs. We first demonstrate how the existing reasoning strategies, Chain-of-Thought and ReAct, can benefit from our structure-oriented analysis. In addition to empirical investigations, we leverage the probabilistic graphical model to theoretically explain why our structure-oriented analysis can improve the LLM reasoning process.

To further improve the reliability in complex question-answering tasks, we propose a multi-agent reasoning system, **S**tructure-oriented **A**utonomous **R**easoning **A**gents (SARA), that can better enforce the reasoning process following our structure-oriented analysis by refinement techniques and is equipped with external knowledge retrieval capability to reduce factual errors. Extensive experiments verify the effectiveness of the proposed reasoning system. Surprisingly, in some cases, the system even surpasses few-shot methods. Finally, the system not only improves reasoning accuracy in complex tasks but also demonstrates robustness against potential attacks that corrupt the reasoning process.

## 1 INTRODUCTION

Large Language Models (LLMs) have shown remarkable potential in various reasoning tasks (Wei et al., 2022; Yao et al., 2022; Shinn et al., 2024; Ahn et al., 2024; Wang et al., 2022), making LLM-based reasoning a fascinating area of research in artificial intelligence. Besides the literature which exhibits LLMs' strong reasoning abilities when provided with task-specific exemplars (Wei et al., 2022; Yao et al., 2022; Besta et al., 2024), more recent studies in zero-shot reasoning methods (Kojima et al., 2022; Qiao et al., 2022) demonstrate their unique advantages. For example, these zero-shot methods explore LLMs' inherent reasoning abilities without human effort in crafting task-specific demonstration examples used in few-shot reasoning and potentially improve the generalization on solving unseen tasks. These benefits highlight the necessity of advancing zero-shot reasoning capabilities in LLMs.

Despite the promising potential of zero-shot reasoning, significant challenges persist. A primary concern is its inferior performance on complex tasks, e.g., answering multi-hop questions, compared to human or few-shot methods (Huang & Chang, 2022; Ahn et al., 2024). Among incorrect responses, it is often observed that zero-shot methods cannot demonstrate human-like thinking processes, such as comprehensively understanding the problem statements.

To address this issue, the concept of human cognition can serve as a valuable reference. Research in human cognition (Simon & Newell, 1971; Kotovsky et al., 1985; Chi et al., 1981; Lakoff & Johnson, 2008) has shown that skilled problem-solvers demonstrate strong reasoning abilities when facing new problems, even without examples or external guidance. They analyze the problem's structure, leveraging linguistic and logical patterns to gain a comprehensive understanding (Lakoff & Johnson, 2008). This analytic thinking process helps identify critical components (Kotovsky et al., 1985) and relationships between these components, extract related sub-questions, and help

identify some key steps along the correct reasoning path. Take the problem in Figure 1 as one example, through understanding the structure of the question, we can obtain the primary objective (identifying a song's name) and its associated constraints (the song's affiliation with a university, and the location of the university's main campus and branches). This analytic thinking process provides a more structured way of reasoning compared to directly exploring the reasoning path.

Inspired by the human analytic thinking process, we introduce a structure-oriented analysis method to improve LLM's zero-shot reasoning capability, which understands the structure of problem statements and generates a comprehensive understanding before performing the reasoning process. The proposed method is based on the syntax and grammar structures in the statement, leveraging LLMs' ability to parse linguistic patterns (Mekala et al., 2022; Ma et al., 2023). With the help of grammar structures, LLMs can accurately identify critical components in the problem statement and relationships among them and further discover related sub-questions. From this perspective, this analytic thinking process mimics human thinking behavior and thus helps explore correct reasoning paths toward solutions. We demonstrate that simply adding this analysis on top of existing methods such as Chain-of-Thought (CoT)(Wei et al., 2022; Kojima et al., 2022) and ReAct (Yao et al., 2022) can significantly enhance the reasoning performance (Section 3.1). Our theoretical analysis (Section 3.2), based on a probabilistic graphical model, also suggests that extracting correct information from problem statements can effectively reduce reasoning errors. All these indicate the potential of our structure-oriented analysis in improving LLMs' inherent reasoning capabilities.

To further boost the effectiveness of our structure-oriented analysis towards solving knowledge-intensive complex problems, we introduce a multi-agent reasoning system, **S**tructure-oriented **A**utonomous **R**easoning **A**gents (SARA), to let the reasoning process better follow the analysis and utilize external knowledge. This system consists of a Reason Agent that generates the structure-oriented analysis; a Refine Agent that evaluates every reason step to check its correctness and alignment with the structure-oriented analysis result; a Retrieve Agent that obtains external knowledge; and a Shared Memory that tracks reasoning trajectories. Our extensive experiments across different tasks and LLMs demonstrate the effectiveness of this system and show that it can achieve comparable or even better performance than few-shot methods (Section 5). Furthermore, we observe enhanced robustness against backdoor attacks (Xiang et al., 2024) and injection attacks (Xu et al., 2024), highlighting additional benefits of our approach in terms of security and reliability.

To summarize, the main scientific contribution of this paper is our observation that the zero-shot reasoning ability of LLMs is not fully explored. Supported by both empirical evidence and theoretical validation, the structure-oriented analysis we propose in this paper significantly enhances the zero-shot reasoning capability of LLMs. Besides the major contribution, an additional contribution is the proposed multi-agent reasoning system, which provides a more comprehensive and practical solution for structure-oriented analysis to further improve the zero-shot reasoning performance.

## 2 RELATED WORK

**LLMs for reasoning.** In literature, there is growing interest in exploring and enhancing the reasoning capability of LLMs. Chain-of-Thought (CoT) prompting, introduced by (Wei et al., 2022), pioneered the approach of encouraging models to generate intermediate reasoning steps, significantly improving the LLMs' performance on multi-step reasoning tasks. Subsequent research has further refined this approach. For instance, (Kojima et al., 2022) proposes zero-shot CoT, which reduces the need for task-specific examples by prompting the model to "think step by step." (Wang et al., 2022) introduces self-consistency to generate multiple reasoning paths and select the most consistent one. Building upon these foundations, several studies have explored more sophisticated reasoning strategies, including exploring more reasoning paths and utilizing feedback to select correct paths. For example, Tree of Thoughts (Yao et al., 2024) characterizes the reasoning process as searching through a combinatorial problem space represented as a tree. Graph of Thoughts (Besta et al., 2024) formulates the reasoning as an arbitrary graph which supports flexible evaluation and refinement for the thoughts. Sub-problem decomposition is also a popular way. (Zhou et al., 2022) directly prompts the LLM to decompose questions into sub-questions with few-shot examples. SOCRATIC CoT(Shridhar et al., 2022) trains a generator to decompose the question and a question-answering model to solve these sub-questions. (Khot et al., 2022) and (Prasad et al., 2023) both proposes to decompose the original problem with a planner or decomposer. Refinement is also a common

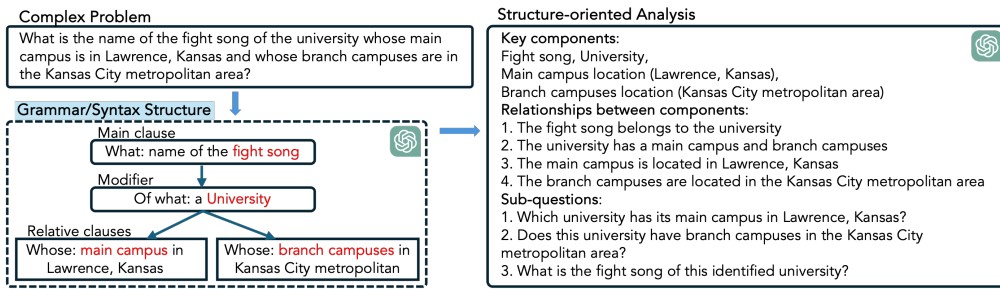

Figure 1: An illustration of the structure-oriented analysis

technique to reduce potential errors. (Shinn et al., 2024; Madaan et al., 2024; Paul et al., 2023) introduce self-reflection, which utilizes the evaluations of LLMs to enhance the correctness of reasoning. (Shridhar et al., 2023b) and (Shridhar et al., 2023a) refine the initial output with another LLM and leverage the third model to select the proper solution. (Zhong et al., 2024) evaluates the most recent reasoning model, OpenAI-o1, and reveals that it takes CoT as a fundamental part of its architecture and leverages it into training to improve the reasoning capability. (Zhou et al., 2024) includes different reasoning strategies and prompts the model to select the proper ones for each question. We notice that most of these methods require task-specific prompting or examples and the zero-shot methods show clear gaps in reasoning performance with few-shot methods. This inspires us to explore the limit of zero-shot reasoning and propose a novel strategy for improvement.

**LLM agents for problem-solving.** Except for the inherent reasoning capability of LLMs, LLM agents are leveraged to further improve the performance of solving complex problems. LLM agents are allowed to digest external feedback and utilize various tools and external knowledge to help the reasoning task. For instance, ReAct (Yao et al., 2022) instructs the model to generate both reasoning traces and task-specific actions in an interleaved manner and allows to gather additional information from external sources. IRCoT (Trivedi et al., 2022) and FreshPrompt (Vu et al., 2023) propose to reinforce the CoT reasoning process by retrieving relevant information. Chain-of-knowledge (Li et al., 2023) proposes dynamic knowledge adapting that can incorporate heterogeneous knowledge sources to reduce factual errors during reasoning. Agent systems specified on different domains are also proposed to boost the performance of corresponding tasks. For instance, MetaGPT (Hong et al., 2023) focuses on software development and breaks complex tasks into subtasks for different agents to work together. Data interpreter (Hong et al., 2024) incorporates external execution tools and logical inconsistency identification in feedback to derive precise reasoning in data analysis tasks. (Zhu et al., 2023) introduces an LLM multi-agent framework including an LLM Decomposer, LLM planner, and LLM interface to conduct tasks and interact with the environment in Minecraft. (Gou et al., 2023b) focuses on tool use of LLMs and trains a series of models with enhanced ability of tool use. (Zhou et al., 2023) proposes an agent system to implement Monte Carlo Tree Search with the help of few-shot examples. (Sumers et al., 2023) summarize the key components of agent systems from the perspective of human cognition and categorize the existing agents. All these works illustrate the power and potential of LLM agents in problem-solving. This inspires us to leverage it to implement our core strategy and fully unleash its power.

## 3 STRUCTURE-ORIENTED ANALYSIS

When skillful human solvers encounter complex questions, a common technique is to first identify the critical components and related sub-questions for a comprehensive understanding of the problem (Kotovsky et al., 1985; Lakoff & Johnson, 2008). This skill can provide a global view of the problem-solving progress, reduce distractions from irrelevant information, and guide for correct reasoning paths (Simon & Newell, 1971). Inspired by these skills, we introduce *structure-oriented analysis*, which leverages LLMs to explicitly extract syntactic and grammatical elements from problem statements to guide the reasoning process.

### 3.1 EMPIRICAL FINDINGS

An example of the structure-oriented analysis can be found in Figure 1. As in the example, we first prompt the LLM to identify the syntactic and grammatical structures of the problem statement,

and then ask the LLM to extract the following key information based on these structures: *key components* that are significant in the problem; *relationships between components* which describe how these critical elements are related in a structured way; *sub-questions* which are smaller and simpler questions that contribute to the final answer. Leveraging LLM's ability in syntax and semantic parsing (Drozdov et al., 2022; Mekala et al., 2022; Ma et al., 2023), we develop a general prompt that is applicable across diverse tasks and problems. This approach reduces the need for task-specific examples, and there is no need for human intervention[1].

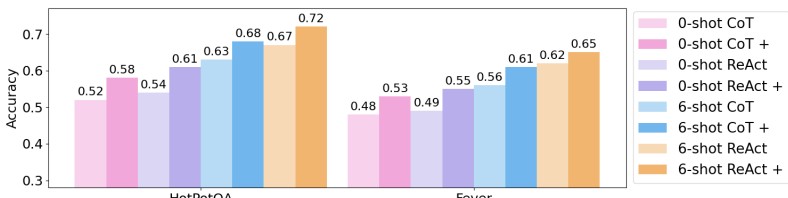

Figure 2: Reasoning accuracy with/without the structure-oriented analysis. The methods with suffixes + are the backbone methods ({CoT, ReAct} × {0-shot, 6-shot}) with structure-oriented analysis added.

To explore the impact of the structured-oriented analysis, we integrate it with two representative reasoning methods–CoT (Wei et al., 2022) and ReAct (Yao et al., 2022), to empirically examine its performance. We consider both 0-shot and 6-shot versions of CoT and ReAct[2]. To be specific, we first prompt the LLM to perform the structure-oriented analysis and let it finish the remaining reasoning process given the analysis. We evaluate the performance of GPT-4 on a multi-hop question answering benchmark HotPotQA (Yang et al., 2018) and a fact verification benchmark Fever (Thorne et al., 2018). Since HotPotQA is a free-form question-answering dataset, a GPT-4 judge is used to compare the output and the ground truth answer. For both tasks, we compare the accuracy with/without our structure-oriented analysis and demonstrate the results in Figure 2. As in Figure 2, adding the structure-oriented analysis can significantly improve the reasoning accuracy, leading to an increase of 5% to 8%. Moreover, compared to 6-shot methods, 0-shot methods gain more improvements. These indicate that without human intervention, LLMs can still have a deeper understanding of the problem with the help of analysis of syntax structures and linguistic patterns, and these understandings further enhance the model's ability to generate more accurate solutions.

## 3.2 THEORETICAL ANALYSIS

Next, we elaborate on how the reasoning happens from a data perspective and understand the potential benefit of our proposed method. Due to the page limit, we provide the skeleton of the analysis and an informal theoretical statement in the main paper and postpone the details to Appendix A.

To briefly introduce the analysis, similar to (Tutunov et al., 2023) and (Xie et al., 2021), we utilize a probabilistic graphical model (PGM) with observed and hidden variables to model the connections among explicit knowledge and abstract concepts in the pre-training data from which LLMs gain reasoning capability. However, unlike previous studies (Prystawski et al., 2024; Tutunov et al., 2023), which assume that the LLM's reasoning process always explores along the correct path in their graphical models, we consider a more general scenario where the LLM may explore an incorrect reasoning path. Our key result shows that identifying the important reasoning steps is crucial in exploring the correct reasoning path.

**Build the PGM.** We use Figure 3 as an example to illustrate the construction of the PGM. The right penal of Figure 3 provides a detailed instance of how the mathematical notations are connected with real data, and the left penal provides a more general case. In the right panel, we denote $\{\theta_i\}_{i=1}^N$ as the *hidden variables* to represent abstract concepts in the data and $\{X_i\}_{i=1}^N$ as the corresponding *observed variables* for pieces of explicit knowledge $\{x_i\}_{i=1}^N$. Here, $\theta_1$ represents the main campuses of universities and their locations, $\theta_2$ can be considered as the locations of branches, $\theta_3$ stands for the tuition fee of the universities, and $\theta_4$ can be interpreted as the fight songs of universities. For each $\theta_i$, the corresponding $X_i$ contains the information of the exact knowledge, such as the location of a specific main campus.

---

[1] Detailed prompt is included in Appendix B

[2] More details can be found in Appendix B

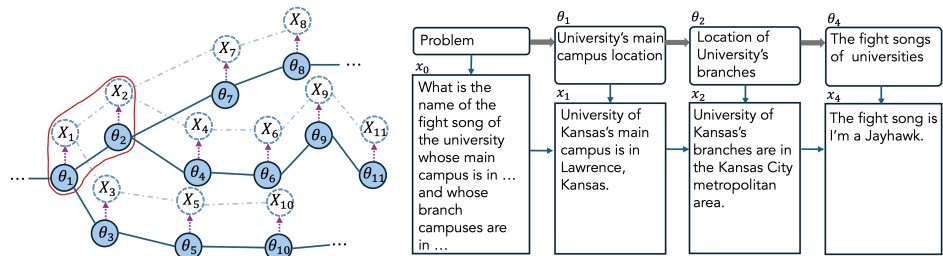

Figure 3: An illustrative example of the PGM generation model. This graph is a part of the underlying PGM where $\theta_i$s are hidden variables and $x_i$s are observed variables. The red circle is an example of the strong connection between $\theta_i$s and $x_i$s in the pre-training.

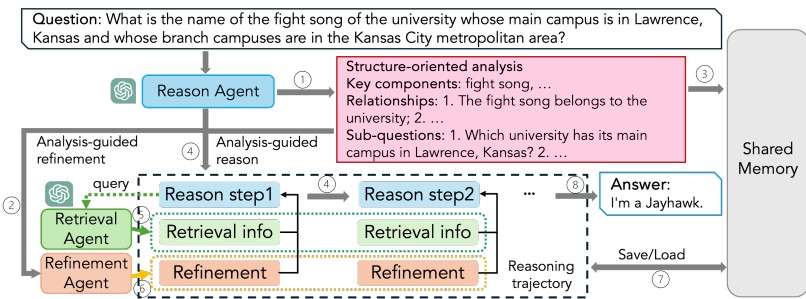

Figure 4: An overview of the Structure-oriented Autonomous Reasoning Agents.

Intuitively, $\theta_1$ (the main campuses of universities and their locations) and $\theta_2$ (the locations of branches) are logically connected. In addition, during the pre-training, LLM can learn the connection between $x_1$ (KU's main campus is in Lawrence, Kansas) and $x_2$ (Kansas City metropolitan area) and similar pairs of $(x_1, x_2)$ for other universities. By leveraging all observed realizations $(x_1, x_2)$ of $(X_1, X_2)$, the LLM can infer the relationship between $\theta_1$ and $\theta_2$. Similarly, the LLM can learn the connection of $(\theta_2, \theta_4)$ in the right panel of Figure 3 to build the PGM.

**Inference.** During the inference stage, to perform reasoning for the example in the right panel of Figure 3, the LLM receives $x_0$ and will explore $\theta_1$ and generate $x_1$. Then, given $(\theta_1, x_1, x_0)$, it will further explore $\theta_2$ and generate $x_2$, etc. In this example, there is a single reasoning chain, $\theta_1 \rightarrow \theta_2 \rightarrow \theta_4$, allowing the LLM to correctly follow the reasoning path.

However, if the PGM learned from pre-training is similar to the left panel of Figure 3, then it may explore an incorrect reasoning path: Suppose the correct final state is $\theta_9$ and the LLM starts the reasoning from $\theta_1$. Since now $\theta_1$ is connected with both $\theta_2$ and $\theta_3$, it can explore either one of them at the inference stage. Furthermore, because the LLM is not pre-trained on the specific test data and does not explicitly perform the same reasoning task, it relies solely on its pre-training knowledge (e.g., pairwise connections among $(\theta_i, \theta_j)$), and may not exactly identify the correct reasoning path.

For our structure-oriented analysis and other similar techniques, as long as the method can identify one or a few correct hidden states for the specific reasoning task and increase the chance of reaching them, then we have the following benefits:

**Theorem 3.1** (Informal Statement of Lemma A.2 and Theorem A.3)**.** *Denote $e(\cdot)$ as the loss given the reasoning path explored by the LLM. Under some mild conditions, if a hidden state $\theta_a$ is in the correct reasoning path, then*

- *P(correct reasoning | $\theta_a$ is explored) $\geq$ P(correct reasoning). The probability of the LLM doing correct reasoning if it can reach $\theta_a$.*

- *$e(\theta_a$ is explored$) \leq e$(LLM randomly explores). The loss, e.g., accuracy or mean square loss, is also smaller if the LLM can reach $\theta_a$ successfully.*

In Appendix A, we provide the rigorous notations and the formal theorem statements.

# 4 AUTONOMOUS REASONING SYSTEM

Although Section 3.1 demonstrates the effectiveness of our structure-oriented analysis, there is still large room for improvement: First, in the experiments of Figure 2, we notice that the LLM cannot always follow the structure-oriented analysis results when performing the reasoning. Second, the LLM sometimes generates inconsistent reasoning results. Finally, some factual errors also occur. Therefore, extra efforts are needed to further unleash the power of our structure-oriented analysis.

Based on the above observations, to obtain a better reasoning capability, an LLM-based question-answering mechanism is desired to be equipped with 1) a design to encourage the reasoning process following the structure-oriented analysis result, 2) consistency in the reasoning trajectory, and 3) the capability of utilizing external knowledge to avoid factual errors. While prompt engineering may incorporate all these expectations into a single prompt, employing multiple agents to modularize the sub-tasks can make the system more robust and general. Therefore, we design a multi-agent reasoning system, **S**tructure-oriented **A**utonomous **R**easoning **A**gents (SARA), with dedicated agents to align the reasoning process with our structure-oriented analysis and ensure the reasoning accuracy through consistency in the reasoning trajectory and addition external knowledge.

## 4.1 SYSTEM DESIGN

SARA consists of four major parts: Reason Agent, Refinement Agent, Retrieval Agent, and Shared Memory. Each agent plays a specific role and cooperates with each other to complete the task.

**Reason Agent.** This agent serves as the cognitive core of the system, conducting analytic thinking and generating detailed reasoning steps. It performs multiple critical functions. Upon receiving a new question, it analyzes the grammar and syntax, which are the rules that determine how words are arranged to form a sentence and generates the structure-oriented analysis. Based on this analysis, it proceeds with a step-by-step reasoning to gradually solve the complex task. Within each step, it is prompted to determine whether external information is needed, and interacts with the Retrieval Agent to obtain external knowledge when necessary. This retrieved knowledge is then incorporated into the subsequent reasoning. After completing the reasoning process, the Reason Agent consolidates a comprehensive final answer based on the entire reasoning trajectory. There is no human intervention needed in this process.

**Refinement Agent.** Prior research has demonstrated that the reasoning capacities of LLMs can be enhanced through refinement processes, including self-refinement (Madaan et al., 2024) and external supervision (Gou et al., 2023a; Shinn et al., 2024). To ensure that the Reason Agent's generated reasoning steps align with the structure-oriented analysis and are free from potential logical errors, we introduce an LLM-driven Refinement Agent. This agent inspects both the structure-oriented analysis and the reasoning trajectory. Specifically, it first examines the structure-oriented analysis to prevent misinterpretations of the problem statement. It then reviews each reasoning step based on the following three criteria: (1) alignment with the structure-oriented analysis, (2) consistency with the previous reasoning trajectory, and (3) factual correctness with relevant external knowledge. This comprehensive inspection is designed to mitigate risks of deviation of the reasoning trajectory from the structure-oriented analysis, resolve inconsistencies or logical errors among reasoning steps, and correct any potential factual inaccuracies based on retrieved knowledge.

**Retrieval Agent.** This agent accesses external knowledge, including pre-constructed databases and web-based resources such as Wikipedia and Google Search. This approach can complement the internal knowledge of LLMs in case the internal knowledge is insufficient, which is determined by the Reason Agent during the reasoning process. Upon receiving a retrieval query from the Reason Agent, the LLM within the Retrieval Agent interprets the request and transforms it into a proper format for the external API/target data resources. By leveraging the relevant external information, the Retrieval Agent enhances the system's reasoning performance by reducing factual errors. Note that the Retrieval Agent only retrieves external knowledge when the Reason Agent identifies missing information and requests it to be retrieved. In this case, we can avoid knowledge conflict since the retrieved knowledge is always a supplement for the Reason Agent.

**Shared Memory.** We utilize a naive Memory module (implemented as a dictionary) to store the structure-oriented analysis result, reasoning trajectory, and retrieved information. The Reason Agent retrieves the structure-oriented analysis result and previous reasoning steps from Shared Memory

and generates new reasoning steps; the Refinement Agent performs the refinement in the context of the structure-oriented analysis result and previous reasoning steps stored in Shared Memory.

## 4.2 REASONING PROCESS

The whole reasoning process of the system is shown in Figure 4. The process consists of three stages: (1) structure-oriented analysis, (2) iterative reasoning, (3) answer consolidation.

**Structure-oriented Analysis**. As discussed in Section 3, effective problem-solving typically begins with a comprehensive understanding of the problem statement. In the enhanced system, when a new question is received, the Reason Agent conducts a thorough analysis (① in Figure 4) based on the syntactic structures of the problem (illustrated in Figure 1). This analysis extracts critical components and generates relevant sub-questions for reference. For instance, in Figure 4 the question asks for the name of the fight song of a university with some constraints on the location of the main campus and branches. The Reason Agent identifies the key components as "fight song, university, main campus,...", and the relationship is that "fight song" is the main objective while it belongs to "university" which is restricted by the location of "main campus". Given these components, some sub-questions can be further derived, e.g., "which university has its main campus located in ...". Besides, to ensure the reasoning accuracy, the initial analysis is sent to the Refinement Agent for evaluation and refinement (② in Figure 4). The Refinement Agent is prompted to provide an explicit reason for its judgments and refinements, which helps mitigate potential hallucinations (Yao et al., 2022). This refined analysis is then stored in the Memory for future reference (③ in Figure 4).

**Iterative reasoning**. To fully harness the reasoning capability of LLMs, we adopt an iterative reasoning strategy (Yao et al., 2022; Wei et al., 2022; Li et al., 2023). As shown in Figure 4, in each iteration, Reason Agent takes the structure-oriented analysis and the previous reasoning trajectory as the context to reason the current step (④ in Figure 4). If external knowledge is needed, the Reason Agent queries the Retrieval Agent (⑤ in Figure 4). The Retrieval Agent then searches for related information from external databases or web data and sends it back to the Reason Agent. For instance, if the current step is "what is the name of the university with the main campus in Lawrence Kansas", the Reason Agent will interact with the Retrieval Agent to obtain "the University of Kansas" from Wikipedia. The Refinement Agent then evaluates and refines this step (⑥ in Figure 4), aligning the step with the structure analysis and its relevance. This evaluation is accompanied by detailed reasons as in ReAct (Yao et al., 2022), enhancing the process's reliability. The refined steps are stored in the Shared Memory for use in subsequent iterations (⑦ in Figure 4) and synchronization of all agents.

**Answer consolidation**. Finally, after the iterative reasoning process, the answer to the original problem is concluded (⑧ in Figure 4).

## 5 EXPERIMENTS

We conduct experiments to verify the effectiveness of the SARA.

## 5.1 EXPERIMENT SETTING

**Agent configurations.** We utilize the same LLM for all LLM-driven agents (Reason Agent, Refinement Agent and Retrieval Agent). Four representative LLMs are tested, including two API-only models, GPT-4 and Qwen-max, and two open-source models, Llama3-70B and Qwen2-57B (Bai et al., 2023). For the Retrieval Agent, if not specified, we use Wikipedia API to obtain external knowledge. SARA is built with the open-source multi-agent framework, AgentScope (Gao et al., 2024), and the detailed prompt templates for each LLM-driven agent are reported in Appendix C.

**Tasks.** We aim to improve the general reasoning capability of LLMs, so we test on various representative reasoning tasks. HotpotQA (Yang et al., 2018) contains multi-hop reasoning questions; Fever (Thorne et al., 2018) is evaluated for fact verification task; MMLU (Hendrycks et al., 2020) is evaluated for multitask language understanding (specifically in Biology and Physics domains, aligning with previous research (Li et al., 2023)); StrategyQA (Geva et al., 2021) evaluates commonsense reasoning ability of models; GSM8K (Cobbe et al., 2021) and MATH (Hendrycks et al., 2021) can test the math reasoning tasks. Among all these tasks, HotpotQA, Fever, MMLU and StrategyQA can take advantage of external knowledge, so we group them as knowledge-intensive tasks. In terms

Table 1: Main results on knowledge-intensive reasoning tasks.

| Models | Tasks | Methods | | | | | | | |
|---|---|---|---|---|---|---|---|---|---|
| | | Vanilla | ICL(6-shot) | CoT(6-shot) | ReAct(6-shot) | CoK(6-shot) | CoT(0-shot) | CoT-SC@10(0-shot) | SARA |
| GPT-4 | HotpotQA | 48.9% | 51.4% | 62.2% | 67.2% | 67.6% | 52.3% | 58.8% | **73.5%** |
| | Fever | 35.3% | 48.4% | 56.1% | 61.7% | 61.3% | 46.9% | 53.1% | **66.2%** |
| | MMLU-BIO | 94.1% | 94.6% | 95.3% | 96.9% | 96.7% | 94.5% | 95.7% | **97.5%** |
| | MMLU-PHY | 65.3% | 66.5% | 69.4% | 74.5% | 73.9% | 66.2% | 68.2% | **78.7%** |
| | StrategyQA | 65.6% | 68.1% | 82.9% | 81.7% | 83.2% | 72.8% | 81.4% | **86.4%** |
| Qwen-max | HotpotQA | 49.6% | 51.7% | 58.3% | 64.7% | 66.3% | 50.6% | 56.7% | **70.2%** |
| | Fever | 29.9% | 39.1% | 48.4% | 58.2% | 53.5% | 41.5% | 50.5% | **63.1%** |
| | MMLU-BIO | 90.2% | 91.3% | 93.4% | 93.9% | 94.1% | 91.6% | 93.5% | **96.2%** |
| | MMLU-PHY | 60.5% | 56.2% | 64.3% | 71.8% | 69.1% | 60.7% | 65.1% | **75.4%** |
| | StrategyQA | 73.4% | 75.5% | 89.6% | 88.4% | 90.5% | 80.4% | 83.1% | **90.7%** |
| Qwen2-57B | HotpotQA | 32.2% | 33.5% | 41.6% | 53.9% | 55.3% | 35.1% | 44.5% | **58.7%** |
| | Fever | 21.5% | 26.3% | 44.7% | 52.6% | 51.3% | 33.2% | 45.6% | **56.1%** |
| | MMLU-BIO | 86.1% | 86.6% | 87.4% | 90.2% | 90.9% | 86.5% | 87.9% | **93.3%** |
| | MMLU-PHY | 53.2% | 55.7% | 63.4% | 66.4% | 68.3% | 56.3% | 63.8% | **71.1%** |
| | StrategyQA | 58.4% | 63.2% | 85.1% | 89.2% | 88.3% | 66.8% | 79.1% | **91.5%** |
| Llama3-70B | HotpotQA | 39.1% | 38.2% | 47.5% | 56.2% | 54.1% | 40.6% | 44.8% | **60.9%** |
| | Fever | 46.4% | 48.5% | 53.1% | 57.7% | 58.2% | 47.3% | 51.9% | **62.8%** |
| | MMLU-BIO | 89.2% | 87.4% | 89.5% | 91.3% | 91.7% | 88.4% | 89.2% | **94.2%** |
| | MMLU-PHY | 47.9% | 48.6% | 55.3% | 61.4% | 60.9% | 49.5% | 55.7% | **65.3%** |
| | StrategyQA | 57.9% | 65.1% | 84.2% | 85.2% | 85.8% | 72.5% | 80.5% | **87.1%** |

Table 2: Main results on math reasoning tasks.

| | Tasks | Methods | | | | | | | |
|---|---|---|---|---|---|---|---|---|---|
| | | Vanilla | ICL(6-shot) | CoT(6-shot) | ReAct(6-shot) | CoK(6-shot) | CoT (0-shot) | CoT-SC@10(0-shot) | SARA |
| GPT4 | GSM8K | 66.8% | 66.9% | 92.1% | 93.7% | 91.9% | 84.3% | 87.8% | **94.2%** |
| | MATH | 43.1% | 55.4% | **69.2%** | 67.5% | 68.6% | 63.6% | 64.1% | 68.2% |
| Qwen-max | GSM8K | 68.6% | 72.8% | 87.5% | 89.2% | 87.6% | 74.8% | 84.2% | **91.3%** |
| | MATH | 42.8% | 45.6% | 64.9% | 64.5% | **65.3%** | 49.3% | 61.9% | 64.7% |
| Qwen2-57B | GSM8K | 54.9% | 59.2% | 82.7% | 83.9% | 83.5% | 63.7% | 74.5% | **84.4%** |
| | MATH | 30.1% | 33.5% | 46.2% | **47.3%** | 46.8% | 31.6% | 40.8% | 46.5% |
| Llama3-70B | GSM8K | 55.3% | 58.3% | 83.7% | 86.5% | 87.2% | 66.5% | 76.8% | **89.7%** |
| | MATH | 30.7% | 32.4% | 42.9% | **46.3%** | 44.9% | 32.8% | 36.4% | 44.2% |

of evaluation metrics, the predicted solutions for HotpotQA and MATH are free-form answers, so we utilize a GPT-4 judge to assess the answer correctness and report the average accuracy as "LLM Acc". For other datasets, we report the average accuracy as "Acc". More details of these datasets are provided in Appendix D.

**Baselines.** We compare SARA with common baselines and some representative reasoning methods: (1) Direct prompting (Vanilla) directly asks the LLM to answer the question. (2) In-context learning (ICL) asks the LLM to solve the problem given examples. (3) (few-shot) Chain-of-thought (CoT (Wei et al., 2022)) prompts the model to generate intermediate steps when solving the problem. (4) ReAct (Yao et al., 2022) combines agent thoughts (reason the current state) and actions (task-specific actions such as Search for an item with Wiki API) to help solve the problem. (5) Chain-of-knowledge (CoK (Li et al., 2023)) uses knowledge from different domains to correct reasoning rationales. Except for the direct prompting, all other baselines use a few-shot prompting strategy, and we test 6-shot as default to align with previous works (Yao et al., 2022; Li et al., 2023). (6) 0-shot CoT (Kojima et al., 2022). (7) 0-shot CoT with self-consistency (Wang et al., 2022) generates multiple CoT solutions and chooses one using a major vote. We generate 10 solutions. Examples of ICL and CoT are randomly selected from the training set for each task; reasoning steps in each CoT example are manually crafted. ReAct and CoK are implemented following the original paper.[3]

## 5.2 MAIN PERFORMANCE ON KNOWLEDGE-INTENSIVE TASKS

The main results of SARA and the baselines on knowledge-intensive tasks are presented in Table 1. In general, SARA consistently outperforms all baselines across all tasks and models used in the experiments. For example, in HotpotQA, compared with baselines without explicit reasoning strategies, such as Vanilla and ICL, SARA achieves significant improvements of over 15% for most tasks. This suggests that even advanced models like GPT-4 and Qwen-max require proper strategies to fully leverage their reasoning capabilities, and simple examples alone are insufficient. To compare SARA with CoT, SARA also substantially improves the reasoning capability and surpasses CoT by over 10%. This superiority can be attributed to three key factors: (1) comprehensive question understanding through our structure-oriented analysis, (2) refinement processes, and (3) integration of

---

[3]Codes available in `https://anonymous.4open.science/r/ReasonAgent-4458`.

external knowledge. In terms of the ReAct and CoK, SARA also demonstrates clear advantages over them with average improvements of 4% and 4.4%, respectively, and the primary difference between these two methods and SARA is our structure-oriented analysis. Moreover, our method outperforms 0-shot CoT SC@10 to a large extent suggesting that structure-oriented analysis can significantly improve the 0-shot reasoning capability. In addition to HotpotQA, SARA also demonstrates significant advantages in other complex reasoning tasks such as HotpotQA, Fever, MMLU-PHY, and MMLU-BIO , highlighting its effectiveness and generalization ability across diverse tasks.

## 5.3 Main performance on math reasoning tasks

In Table 1, we present the main results of math reasoning tasks. Among all datasets, few-shot baselines significantly outperform 0-shot baselines, indicating a significant performance gap between few-shot and 0-shot reasoning capability. Our method consistently outperforms 0-shot baselines and even works better than few-shot baselines on the GSM8K dataset. This shows that structure analysis can generalize well to math reasoning tasks especially when the problem is described in natural language such as GSM8K. We do notice that SARA is not the best on the MATH dataset. This can be because some MATH problems are expressed in symbols, which do not have clear structures for analysis. Our method can still have comparably good results on MATH suggesting the benefit of step-wise reasoning and refinement in the agent design.

## 5.4 Effect of structure-oriented analysis

To elucidate the impact of the structure-oriented analysis, we conduct experiments evaluating the effectiveness of the three crucial functions in the Reason Agent: (1) key components and relationships between components, (2) sub-questions, and (3) grammar/syntax structure. Using GPT-4 on all reasoning tasks, we test different combinations of these elements, as detailed in Table 3[4].

There are several observations from Table 3. Consider HotpotQA as an example. First, comparing Settings 1, 2, and 3, when the grammar/syntax structure is included, removing either key components (Setting 2) or sub-questions (Setting 3) has only a small decrease in the performance. However, in Setting 4, excluding the grammar/syntax structure significantly reduces performance by over 10%, suggesting the importance of the grammar/syntax structure. Second, comparing Setting (1, 3) and (5, 7), without the key components and grammar/syntax structure analysis, formulating sub-questions only has limited improvement of 1.9% on the reasoning performance, lower than 4.1% in Setting (1, 3). Similar observations can be found in Settings (1,2) and (6,7) for the key components, which indicates the synergy effect of grammar/syntax with key components and sub-questions. Third, completely removing the structure-oriented analysis also substantially diminishes reasoning performance. The above observations are consistent across all tasks considered.

## 5.5 Effect of key agents

In this subsection, we study the effect of two key agents in SARA, the Refinement Agent and the Retrieve Agent. We test with GPT-4 model on Hot-potQA and Fever benchmarks and summarize the results in Figure 6. When replacing the original LLM (GPT-4) with a smaller model (Qwen2-57) in the Retrieval Agent, the performance is barely affected; while for the Refine Agent, the performance drops a bit more. This suggests that it is feasible to utilize a smaller model in the Retrieval Agent for efficiency while maintaining effectiveness, but the Refine Agent requires strong models.It is noted that re-

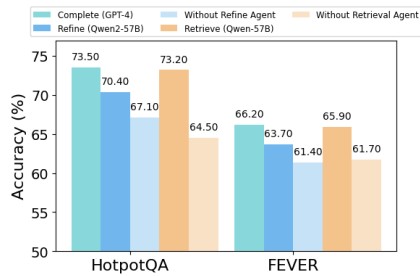

Figure 5: Ablation study on agents. Refinement Agent and Retrieval Agent are removed and reasoning performance is tested respectively.

moving either agent will decrease the reasoning capacity of the system. Moreover, without the Refinement Agent, SARA still has a comparable performance with ReAct and CoK (Table 1), and without the Retrieval Agent, SARA can also achieve better results than 6-shot CoT (no retrieval as well). These highlight the effectiveness of structure-oriented analysis.

---

[4]Since grammar/syntax is used for extracting key components and sub-questions, we do not consider the case only grammar/syntax is removed.

Table 3: Effect of each component in the reasoning agent. 'O' means include and 'X' means exclude.

| Setting # | 1 | 2 | 3 | 4 | 5 | 6 | 7 |
|---|---|---|---|---|---|---|---|
| **Key components** | O | X | O | O | X | O | X |
| **Sub-questions** | O | O | X | O | O | X | X |
| **Grammar/syntax** | O | O | O | X | X | X | X |
| **HotpotQA** | 73.5% | 69.2% | 69.4% | 59.6% | 58.6% | 58.1% | 56.5% |
| **Fever** | 66.2% | 61.7% | 62.1% | 53.4% | 53.1% | 52.9% | 52.3% |
| **MMLU-bio** | 97.5% | 96.3% | 96.6% | 94.1% | 94.3% | 94.1% | 93.9% |
| **MMLU-phy** | 78.7% | 74.1% | 74.6% | 59.5% | 59.1% | 57.2% | 57.6% |

## 5.6 EVALUATION OF ROBUSTNESS

Table 4: Robustness evaluation, accuracy on GPT-4 after attack. Clean accuracy is included in brackets.

| Attack | Task | Vanilla | ICL(6-shot) | CoT(6-shot) | ReAct(6-shot) | CoK(6-shot) | SARA |
|---|---|---|---|---|---|---|---|
| **Badchain** | **HotpotQA** | 48.4%(48.9%) | 13.7%(51.4%) | 14.1%(62.2%) | 21.3%(67.2%) | 16.7% (67.6%) | 71.3% (73.5%) |
| | **Fever** | 35.5%(35.3%) | 25.3% (48.4%) | 12.1% (56.1%) | 10.8% (61.7%) | 21.8%(61.3%) | 64.9% (66.2%) |
| **Preemptive attack** | **HotpotQA** | 33.5% (48.9%) | 42.1% (51.4%) | 41.6% (62.2%) | 55.3% (67.2%) | 56.1% (67.6%) | 68.2%(73.5%) |
| | **Fever** | 19.2%(35.3%) | 39.6%(48.4%) | 32.2%(56.1%) | 54.2%(61.7%) | 52.3%(61.3%) | 61.9%(66.2%) |

Despite the improvement in the reasoning capability, we surprisingly find that SARA is robust to potential corruptions or distractions that target the reasoning process (Xiang et al., 2024; Xu et al., 2024). We evaluate the robustness of SARA against two attacks: BadChain (Xiang et al., 2024)targeting few-shot reasoning methods, which inserts backdoor reasoning steps during the model's reasoning process through poisoned demonstrations; and Preemptive Attack (Xu et al., 2024) targeting 0-shot methods, which inserts a malicious answer directly into the query to mislead the reasoning process. We test both attacks on HotpotQA and Fever with GPT-4, and the results are summarized in Table 4 [5]. When applying Badchain to our method, we simply replace the original input with input attached to the trigger. While few-shot baselines show high vulnerability to BadChain and Vanilla prompting performs poorly under Preemptive Attack, SARA effectively resists both types of attacks. The robustness of SARA can be attributed to two factors: (1) SARA's zero-shot nature, which prevents malicious injections in demonstrations, and (2) the structure-oriented analysis, which focuses on syntax and grammar structures and therefore filters out irrelevant information in problem.

## 6 CONCLUSION

In this paper, inspired by human cognition, we introduce structure-oriented analysis to encourage LLMs to understand the query in a more formulated way. Utilizing the analysis result, LLMs can better identify key steps when performing reasoning tasks, improving reasoning performance. Furthermore, built upon the structure-oriented analysis, we further establish a multi-agent reasoning system to comprehensively improve the consistency and reliability of the LLM's reasoning process. Since this paper mainly focuses on knowledge-intensive tasks, future works can explore other types of tasks, such as mathematical reasoning.

## 7 LIMITATION

Although our strategy shows effectiveness on diverse reasoning tasks, including knowledge-intensive reasoning, math reasoning, and commonsense reasoning, we notice that our method works better on problems that are clearly described in natural languages, such as GSM8K, while performs worse on pure symbol expressions as no obvious structures appear like some questions in MATH dataset. This suggests a future direction for extracting logic structures and learning symbolic expressions to improve reasoning capability. Besides, the LLM agent we adopt to illustrate our principal strategy is simple to fit in various tasks, which can still have room for improvement. Modifying the agent system while maintaining the core structure analysis to adapt to different tasks can be a potential direction. For example, when solving math problems, instead of the Retrieve Agent, leveraging external tools like a calculator or code executor to improve the performance.

---

[5]Experimental details are provided in Appendix E

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

# A  THEORETICAL ANALYSIS

## A.1  THEORETICAL ANALYSIS

In addition to the PGM introduced in Section 3.2, we provide more details on our assumption in the LLM and the notations of the reasoning path. Then we provide a formal statement of Theorem 3.1.

**LLM in pretraining.** Recall that in Figure 3, the PGM contains hidden variables $\{\theta_i\}_{i=1}^N$ as the observed variables $\{X_i\}_{i=1}^N$ with the explicit knowledge $\{x_i\}_{i=1}^N$. Following a similar idea as in (Prystawski et al., 2024), when using the above pre-training data to train an LLM $\mathcal{M}$, the output of $\mathcal{M}$ satisfies the following properties. First, most existing LLMs used for complex tasks demonstrate reliable capability in telling whether two given pieces of explicit knowledge share the same abstract concept or not (i.e., whether $x_i$ and $x'_j$ share the same $\theta$). Based on this, we assume that the LLMs can faithfully capture the relationship between the hidden variables and the corresponding explicit knowledge (i.e., the edges between $\theta_i$ and $X_i$). Moreover, since most LLMs are trained for next-token prediction, explicit knowledge and abstract concepts that frequently appear in nearby within texts (i.e., the connections between $x_i$ and $x_j$ as well as the connection between $\theta_i$ and $\theta_j$) are also learned by LLMs with high quality. For example, information about the main campus of the University of Kansas and its branches often appears within the same paragraph on a Wikipedia page; generally, the location of universities and their branches locations usually appear close in text.

**Use PGM to explain the reasoning process.** In Section 3.2, we intuitively explain the reasoning process using the examples in Figure 3. The detailed mathematical description of the reasoning procedure is as follows. The model $\mathcal{M}$ receives an input question $x_0$, e.g., "find the name of the fight song of the university whose main campus is in ..." in the right panel of Figure 3, and the target is to infer the answer via exploring different variables in the PGM. Define a *reasoning path* $\gamma$ as a set of indexes $\{s_i\}$ of hidden and observed variables $(\theta_{s_i}, x_{s_i})$. The *correct reasoning path* $\gamma^*$ is an ideal reasoning path that both logically correct and leading to the final correct answer. As for the example in Figure 3, the correct reasoning path is $\gamma^* := 1 \to 2 \to 4$, i.e., exploring through hidden states $\theta_1 \to \theta_2 \to \theta_4$. *Ideally*, if $\mathcal{M}$ follows $\gamma^*$, it will output $x_1|x_2|x_4$. However, because the abstract concepts and explicit knowledge in multi-hop reasoning of a complex question are unlikely to appear in pre-training data all close to each other, $\mathcal{M}$ has no direct knowledge of $\gamma^*$ but can only focus on the next variable exploration based on the edges in PGM when reasoning. As a result, instead of the correct reasoning path $\gamma^*$, we assume that $\mathcal{M}$ explores actual reasoning path step by step: given $s_i$ and $x_{s_i}$, $\mathcal{M}$ explores $\theta_{s_{i+1}}$ and generates $x_{s_{i+1}}$ from $X_{s_{i+1}}|x_{s_i}, \theta_{s_{i+1}}$, and all the explored $s_i$s together form the reasoning path $\gamma$. The $\gamma$ also involves randomness since $\mathcal{M}$ is a generation model. Finally, to ease the later analysis, denote $\Gamma(x_0, \cdot, \mathcal{M})$ and $\Gamma(x_0, \theta_T, \mathcal{M})$ as the set of all possible reasoning paths and the set of all *correct* paths respectively, where $\theta_T$ is the correct final reasoning step (the target).

In the following, we analyze how additional information about intermediate variables lying on the correct reasoning path benefits multi-step reasoning.

**Quantify the benefit of correct intermediate variables.** Given $x_0$, we denote $\mathcal{E}(\gamma)$ as *reasoning error* for a given reasoning path $\gamma$ to quantify the performance and $e(\Gamma) \triangleq \sum_{\gamma \in \Gamma} P(\gamma)\mathcal{E}(\gamma)$ as the *expected reasoning error* for a set of paths $\Gamma$, and study how the choice of $\Gamma$ affects $e(\Gamma)$.

When performing the reasoning with the structure-oriented analysis, the analysis can extract a sequence of indices of latent variables $A = \{s_1^A, s_2^A, \ldots\}$, which can be key components or sub-questions in practice as shown in Figure 1. In the following, we first provide some mild assumptions on $\gamma$, and then demonstrate how the reasoning error is impacted by $A$.

**Assumption A.1.** Given $x_0$, the random variable $\gamma$ satisfies the following conditions: (1) $\Gamma(x_0, \theta_T, \mathcal{M})$ contains only one path: $\Gamma(x_0, \theta_T, \mathcal{M}) = \{\gamma^*\}$. (2) $\mathcal{E}(\gamma) \geq 0$ and equals to 0 iff $\gamma = \gamma^*$.

In Assumption A.1, the first condition in Assumption A.1 assumes a unique correct path. Discussion for a relaxed version for multiple correct paths can be found in Remark A.4. In the second condition, the reasoning error is zero only when we explore the correct path.

Given the above notations and assumptions, the following result holds:

**Lemma A.2.** *Let $\Gamma_A(x_0, \cdot, \mathcal{M})$ denote the set of explored paths given $A$. Under Assumption A.1, assume that $A \subseteq \gamma^*$, then the following results in $\theta_T$ (with the corresponding index $T$) and $\gamma$ hold:*

*(1) When $|A| = 1$, i.e. $A = \{s^A\}$ for some $s^A \in \gamma^*$, then $P(T \in \gamma|s^A \in \gamma) \geq P(T \in \gamma)$ where the equality holds if and only if $P(s^A \in \gamma) = 1$.*

*(2) When $|A| > 1$, i.e. $A = \{s_1^A, \ldots, s_k^A\}$, and $A \subseteq \gamma^*$, we have a sequence of inequalities*

$$P(T \in \gamma|A \subseteq \gamma) \geq P(T \in \gamma|\{s_j^A\}_{j \in [k-1]} \subseteq \gamma) \geq \ldots \geq P(T \in \gamma).$$

The proof of Lemma A.2 can be found in Appendix A.2. Based on Lemma A.2, when the LLM follows $A$ and explores the variables $\{s_j^A\}_{j \in [k]}$, there is a higher chance that it finally explores $\theta_T$.

Besides the probability of reaching $\theta_T$ considered in Lemma A.2, the following theorem presents the results on how the expected reasoning error is impacted by $A$. We consider two specific errors: (1) 0-1 error $\mathcal{E}_{0-1}(\gamma) = \mathbf{1}(T \notin \gamma)$, and (2) the probability error considered in (Prystawski et al., 2024)

$$\mathcal{E}_{\text{prob}}(\gamma) = \mathbb{E}_{\{(X_i,\theta_i)\}_{i \in G}} \left[ p(X_T = x_t|x_0, \{(X_i, \theta_i)\}_{i \in \gamma}) - p(X_T = x_t|x_0, \{(X_i, \theta_i)\}_{i \in G}) \right]^2$$

with $G$ as all variables in the PGM. We quantify the expected reasoning error as follows:

**Theorem A.3.** *Under the assumptions in Lemma A.2, for $\mathcal{E} \in \{\mathcal{E}_{0-1}, \mathcal{E}_{prob}\}$, the following holds:*

*(1) When $|A| = 1$, i.e. $A = \{s^A\}$ for some $s^A \in \gamma^*$,*

$$e(\Gamma_A(x_0, \cdot, \mathcal{M})) \leq e(\Gamma(x_0, \cdot, \mathcal{M}))$$

*where the equality holds only if $P(s^A \in \gamma) = 1$.*

*(2) When $|A| > 1$, i.e. $A = \{s_1^A, \ldots, s_k^A\}$, and $A \subseteq \gamma^*$, we have a sequence of inequalities*

$$e(\Gamma_A(x_0, \cdot, \mathcal{M})) \leq e(\Gamma_{\{s_j^A\}_{j \in [k-1]}}(x_0, \cdot, \mathcal{M})) \leq \ldots \leq e(\Gamma(x_0, \cdot, \mathcal{M})).$$

The proof of Theorem A.3 can be found in Appendix A.2. Theorem A.3 implies that given the information of the variables on the correct path, the reasoning error is reduced.

*Remark* A.4 (Multiple correct paths). Though Assumptions A.1 assumes a unique correct path $\gamma^*$, it is possible that there exist multiple correct paths in practice. The above result also holds when multiple correct paths exist given some mild conditions on $A$. Suppose there exist multiple correct paths, i.e. $\Gamma^* = \{\gamma_1^*, \gamma_2^*, \ldots\}$, and we assume that $\mathcal{E}(\gamma_i^*) = 0$ for these reasoning paths. We still consider a sequence of indices of latent variables $A = \{s_1^A, s_2^A, \ldots\}$ lying on these correct paths. In particular, we assume there is a subset $A^*$, such that every index in $A^*$ lies on every correct path, denoted as $A^* \subseteq \Gamma^*$. Then the results in Theorem A.3 still hold by replacing $A$ with $A^*$ and $\gamma^*$ with $\Gamma^*$. This is because errors of paths out of $\Gamma^*$ are all positive, and information of $A^*$ significantly increases the probability of inferring paths in $\Gamma^*$ and thus decreases the reasoning error.

*Remark* A.5 (Error when the exploration is not guaranteed to find $\theta_s$ for some $s \in A$). In practice, when searching a proper reasoning path, it is possible that the exploration does not guarantee to reach $\theta_s$ for $s \in A$ for sure. Assume $|A| = 1$. In this case, denote $\Gamma \backslash \Gamma_A$ as the reasoning path that does not pass $A$, and then the total error becomes

$$P(\theta_s \text{ is reached})e(\Gamma_s(x_0, \cdot, \mathcal{M})) + P(\theta_s \text{ is not reached})e(\Gamma \backslash \Gamma_A(x_0, \cdot, \mathcal{M})),$$

and for $\mathcal{E}_{0-1}$ and $\mathcal{E}_{\text{prob}}$, $e(\Gamma \backslash \Gamma_A(x_0, \cdot, \mathcal{M})) \geq e(\Gamma_A(x_0, \cdot, \mathcal{M}))$ as long as the exploration reaches $s$ with a higher chance than random search.

## A.2 PROOFS 3

### A.2.1 PROOF OF LEMMA A.2

*Proof of Lemma A.2.* The proof of Lemma A.2 mainly utilizes the definition of conditional probability. We start from the simple case where $|A| = 1$.

***Single variable in*** *A*. When $A = \{s^A\}$, i.e., only a single variable in $A$, we have

$$P(T \in \gamma) = P(T \in \gamma | s^A \in \gamma) \underbrace{P(s^A \in \gamma)}_{\leq 1} + \underbrace{P(T \in \gamma | s^A \notin \gamma)}_{=0} P(s^A \notin \gamma) \leq P(T \in \gamma | s^A \in \gamma).$$

***Multiple variables in*** *A*. When there are multiple variables in $A$, i.e. $s_1^A, s_2^A, \ldots, s_k^A$, repeat the above analysis, we have

$$P(T \in \gamma) = P(T \in \gamma | A \subseteq \gamma)P(A \subseteq \gamma) + \underbrace{P(T \in \gamma | A \subsetneq \gamma)}_{=0} P(A \subsetneq \gamma) = P(T \in \gamma | A \subseteq \gamma)P(A \subseteq \gamma).$$

Furthermore, it is easy to see that $P(\cap_{j=1}^{i+1}\{s_j^A \in A\}) \leq P(\cap_{j=1}^{i}\{s_j^A \in A\})$, which implies that

$$P(T \in \gamma | \{s_j^A\}_{j \in [i+1]}) \geq P(T \in \gamma | \{s_j^A\}_{j \in [i]})$$

Then we have a sequence of inequalities

$$P(T \in \gamma | A \subseteq \gamma) \geq P(T \in \gamma | \{s_j^A\}_{j \in [k-1]} \subseteq \gamma) \geq \ldots \geq P(T \in \gamma)$$

which completes the proof. $\qquad\qquad\square$

### A.2.2 EXPECTED REASONING LOSS WITH SPECIFIC ERROR FUNCTIONS

We discuss two representative error functions, 0-1 error and probability error, in Theorem A.3.

**0-1 error.** Recall that for a given reasoning path $\gamma$, we define 0-1 error function as

$$\mathcal{E}(\gamma) = \mathbf{1}(T \notin \gamma),$$

where $T$ represents the index of the target variable. This function assigns an error of 0 when the reasoning path reaches the target variable, and 1 otherwise. This binary error metric is both practical and commonly used in evaluating reasoning performance, as it focuses on the logical correctness of the reasoning process. It closely relates to popular empirical metrics such as exact match (EM) (hug, 2023).

*Proof of Theorem A.3, 0-1 error.* Given the above definition of 0-1 error, we have

$$e(\Gamma(x_0, \cdot, \mathcal{M})) = \sum \mathcal{E}(\gamma)P(\gamma) = \sum_{T \notin \gamma} P(\gamma) = P(T \notin \gamma),$$

and

$$e(\Gamma_A(x_0, \cdot, \mathcal{M})) = \sum_{T \notin \gamma} P(\gamma | A \subseteq \gamma) = P(T \notin \gamma | A \subseteq \gamma),$$

both of which are reduced to the probability of $T$ being reached by the reasoning process. As a result, following Lemma A.2, we have $e(\Gamma(x_0, \cdot, \mathcal{M})) \geq e(\Gamma_A(x_0, \cdot, \mathcal{M}))$.

Furthermore, given that $P(T \in \gamma | A \subseteq \gamma) = P(T \in \gamma)/P(A \subseteq \gamma)$, a decrease in $P(A \subseteq \gamma)$ leads to an increase in the improvement gained by conditioning on $A$. This implies that for more complex problems where inferring critical steps in $A$ is challenging, extracting information of $A$ through analysis becomes increasingly important. Following the steps in Lemma A.2, we also have

$$e(\Gamma_A(x_0, \cdot, \mathcal{M})) \leq e(\Gamma_{\{s_j^A\}_{j \in [k-1]}}(x_0, \cdot, \mathcal{M})) \leq \ldots \leq e(\Gamma(x_0, \cdot, \mathcal{M})).$$

$$\square$$

**Probability error.** Recall that the probability error is defined as

$$\mathcal{E}(\gamma) = \mathbb{E}_{\{(X_i, \theta_i)\}} \left[ p(X_T = x_t | x_0, \{(X_i, \theta_i)\}_{i \in \gamma}) - p(X_T = x_t | x_0, \{(X_i, \theta_i)\}_{i \in G}) \right]^2.$$

where $x_t$ is the ground truth output for the target step. The first term is the probability of predicting ground truth given path $\gamma$ while the second term is the probability of predicting the ground truth given the underlying PGM. This error is connected with the widely used cross-entropy loss (Prystawski et al., 2024).

The following lemma presents a valid decomposition of the probability error. Denote $G \backslash \gamma$ as the set of indexes in all paths excluding $\gamma$.

**Lemma A.6** (Decomposition of probability error.). *The following decomposition holds:*

$$\mathcal{E}(\gamma)$$
$$= \mathbb{E}_{\{(X_i,\theta_i)\}_{i\in\gamma}} \mathbb{E}_{\{(X_i,\theta_i)\}_{i\in G\setminus\gamma}} [p(X_T = x_t|x_0, \{(X_i,\theta_i)\}_{i\in\gamma}) - p(X_T = x_t|x_0, \{(X_i,\theta_i)\}_{i\in G})]^2$$
$$= \mathbb{E}_{\{(X_i,\theta_i)\}_{i\in\gamma}} \left[ p(X_T = x_t|x_0, \{(X_i,\theta_i)\}_{i\in\gamma}) - \mathbb{E}_{\{(X_i,\theta_i)\}_{i\in G\setminus\gamma}} p(X_T = x_t|x_0, \{(X_i,\theta_i)\}_{i\in G}) \right]^2$$
$$+ \mathbb{E}_{\{(X_i,\theta_i)\}_{i\in\gamma}} \mathbb{E}_{\{(X_i,\theta_i)\}_{i\in G\setminus\gamma}} \big[ p(X_T = x_t|x_0, \{(X_i,\theta_i)\}_{i\in G}) -$$
$$\mathbb{E}_{\{(X_i,\theta_i)\}_{i\in G\setminus\gamma}} p(X_T = x_t|x_0, \{(X_i,\theta_i)\}_{i\in G}) \big]^2$$

*When $\gamma = \gamma^*$,*

$$\mathcal{E}(\gamma) = 0.$$

The decomposition in Lemma A.6 consists of two parts, where the first part represents the bias of prediction for a given path $\gamma$ while the second term represents the variance.

Given the above decomposition, below is the proof of Theorem A.3 for the probability error:

*Proof of Theorem A.3, probability error.* Similar to the proof of Lemma A.2, we start from the simple case where $|A| = 1$.

***Simple variable in*** $A$. If the model $\mathcal{M}$ can always explore a path with an intermediate variable $\theta_{s^A}$ lying in the correct reasoning path $\gamma^*$, then

$$e(\Gamma_A(x_0, \cdot, \mathcal{M}))$$
$$= \sum_{T \notin \gamma, \gamma \in \Gamma_A(x_0, \cdot, \mathcal{M})} P(\gamma|s^A \in \gamma)\mathcal{E}(\gamma) + \sum_{T \in \gamma, \gamma \in \Gamma_A(x_0, \cdot, \mathcal{M})} P(\gamma|s^A \in \gamma)\mathcal{E}(\gamma)$$
$$= \sum_{T \notin \gamma} \frac{P(\gamma, s^A \in \gamma)}{P(s^A \in \gamma)}\mathcal{E}(\gamma) + \sum_{T \in \gamma} \frac{P(\gamma, s^A \in \gamma)}{P(s^A \in \gamma)}\mathcal{E}(\gamma)$$
$$= \sum_{T \notin \gamma} \frac{P(\gamma, s^A \in \gamma)}{P(s^A \in \gamma)}\mathcal{E}(\gamma).$$

Now we look at the different values of $\mathcal{E}(\gamma)$ when changing $\gamma$. Note that from how the PGM is constructed, we have

$$p(X_T = x_t|x_0, \{(X_i,\theta_i)\}_{i\in\gamma}) = p(X_T = x_t|x_0, \{(X_i,\theta_i)\}_{i\in\gamma^*\cap\gamma}),$$

and

$$p(X_T = x_t|x_0, \{(X_i,\theta_i)\}_{i\in G}) = p(X_T = x_t|x_0, \{(X_i,\theta_i)\}_{i\in\gamma^*}).$$

For any two reasoning paths $\gamma_1$ and $\gamma_2$ so that $s^A \notin \gamma_1$ but $s^A \in \gamma_2$, following similar decompositions as in Lemma A.6, we have

$$\mathcal{E}(\gamma_1)$$
$$= \mathbb{E}_{\{(X_i,\theta_i)\}_{i\in\gamma\cap\gamma^*}} \mathbb{E}_{\{(X_i,\theta_i)\}_{i\in\gamma^*\cap(\gamma_2\setminus\gamma_1)}} \mathbb{E}_{\{(X_i,\theta_i)\}_{i\in\gamma^*\setminus\gamma_2}}$$
$$[p(X_T = x_t|x_0, \{(X_i,\theta_i)\}_{i\in\gamma_1\cap\gamma^*}) - p(X_T = x_t|x_0, \{(X_i,\theta_i)\}_{i\in\gamma^*})]^2$$
$$= \mathbb{E}_{\{(X_i,\theta_i)\}_{i\in\gamma\cap\gamma^*}} \mathbb{E}_{\{(X_i,\theta_i)\}_{i\in\gamma^*\cap(\gamma_2\setminus\gamma_1)}} \mathbb{E}_{\{(X_i,\theta_i)\}_{i\in\gamma^*\setminus\gamma_2}}$$
$$[p(X_T = x_t|x_0, \{(X_i,\theta_i)\}_{i\in\gamma_1\cap\gamma^*}) - p(X_T = x_t|x_0, \{(X_i,\theta_i)\}_{i\in\gamma_2\cap\gamma^*})$$
$$+ p(X_T = x_t|x_0, \{(X_i,\theta_i)\}_{i\in\gamma_2\cap\gamma^*}) - p(X_T = x_t|x_0, \{(X_i,\theta_i)\}_{i\in\gamma^*})]^2$$
$$= \mathbb{E}_{\{(X_i,\theta_i)\}_{i\in\gamma\cap\gamma^*}} \mathbb{E}_{\{(X_i,\theta_i)\}_{i\in\gamma^*\cap(\gamma_2\setminus\gamma_1)}} \mathbb{E}_{\{(X_i,\theta_i)\}_{i\in\gamma^*\setminus\gamma_2}}$$
$$[p(X_T = x_t|x_0, \{(X_i,\theta_i)\}_{i\in\gamma_1\cap\gamma^*}) - p(X_T = x_t|x_0, \{(X_i,\theta_i)\}_{i\in\gamma_2\cap\gamma^*})]^2$$
$$+ \mathbb{E}_{\{(X_i,\theta_i)\}_{i\in\gamma\cap\gamma^*}} \mathbb{E}_{\{(X_i,\theta_i)\}_{i\in\gamma^*\cap(\gamma_2\setminus\gamma_1)}}$$
$$[p(X_T = x_t|x_0, \{(X_i,\theta_i)\}_{i\in\gamma_2\cap\gamma^*}) - p(X_T = x_t|x_0, \{(X_i,\theta_i)\}_{i\in\gamma^*})]^2$$

$$\geq \quad \mathbb{E}_{\{(X_i,\theta_i)\}_{i\in\gamma\cap\gamma^*}} \mathbb{E}_{\{(X_i,\theta_i)\}_{i\in\gamma^*\cap(\gamma_2\setminus\gamma_1)}}$$

$$\left[p(X_T = x_t|x_0, \{(X_i,\theta_i)\}_{i\in\gamma_2\cap\gamma^*}) - p(X_T = x_t|x_0, \{(X_i,\theta_i)\}_{i\in\gamma^*})\right]^2$$

$$= \quad \mathcal{E}(\gamma_2),$$

from which it is easy to see that

$$e(\Gamma(x_0,\cdot,\mathcal{M})) \geq e(\Gamma_A(x_0,\cdot,\mathcal{M})).$$

***Multiple variables in*** $A$. When $|A| > 1$, the steps are indeed the same as when $|A| = 1$. We prove the relationship between $\mathcal{E}(\gamma_1) \geq \mathcal{E}(\gamma_2)$ for different $s_i^A$s.

$\square$

## B   DETAILS FOR EXPERIMENTS IN SECTION 3

**Prompt for structure-oriented analysis.** To add the structure-oriented analysis on top of the backbone reasoning method, we develop the following prompt to let the model identify critical components, relationships among them, and related sub-questions. The LLM is also prompted to provide justification for its analysis.

```
structure-oriented analysis
You are a helpful assistant good at parsing the syntax and grammar
structure of sentences.  Please first analyze the syntax and
grammar structure of the problem and provide a thorough analysis
by addressing the following tasks:
1.Identify Key Components:  Identify the crucial elements and
variables that play a significant role in this problem.
2.Relationship between Components:  Explain how the key components
are related to each other in a structured way.
3.Sub-Question Decomposition:  Break down the problem into the
following sub-questions, each focusing on a specific aspect
necessary for understanding the solution.
4.Implications for Solving the Problem:  For each sub-question,
describe how solving it helps address the main problem.  Connect
the insights from these sub-questions to the overall strategy
needed to solve the main problem.
Question:
```

**Examples for CoT.** For 0-shot CoT, we use the simple prompt "Please think step by step" as in (Kojima et al., 2022). For 6-shot CoT, we manually craft examples for randomly selected problems. It is worth noting that when we add structure-oriented analysis to 6-shot CoT, we simply add it before the standard CoT prompt (Wei et al., 2022). Therefore, in the examples, we still use the original problem rather than the generated analysis. We present some examples as follows.

```
HotpotQA
You need to solve a problem.  Please think step-by-step.  Please
provide your thoughts and then give the final answer.  Thought can
reason about the problem.  Answer can conclude the final answer.

Here are some examples.
Question:  Musician and satirist Allie Goertz wrote a song about
the ̈The Simpsons ̈character Milhouse, who Matt Groening named after
who?
Thought:  Let's think step by step.  Milhouse was named after U.S.
president Richard Nixon, so the answer is Richard Nixon.
Answer:  Richard Nixon

Here are some examples.
Question:  Musician and satirist Allie Goertz wrote a song about
the ̈The Simpsons ̈character Milhouse, who Matt Groening named after
```

who?
Thought: Let's think step by step. Milhouse was named after U.S.
president Richard Nixon, so the answer is Richard Nixon.
Answer: Richard Nixon

Question: Guitars for Wounded Warriors is an album that was
recorded in the village in which New York county?
Thought: Let's think step by step. Guitars for Wounded Warriors
was recorded at Tarquin's Jungle Room Studios in New Paltz
(village), New York. New Paltz is a village in Ulster County
located in the U.S. state of New York. So the answer is Ulster
County.
Answer: Ulster County
...

Fever
Determine if there is Observation that SUPPORTS or REFUTES a
Claim, or if there is NOT ENOUGH INFORMATION. Please think step
by step. Here are some examples.
Claim: Nikolaj Coster-Waldau worked with the Fox Broadcasting
Company.
Answer: Let's think step by step. Nikolaj William Coster-Waldau
appeared in the 2009 Fox television film Virtuality, so he has
worked with the Fox Broadcasting Company. So the answer is
SUPPORTS

Claim: Stranger Things is set in Bloomington, Indiana.
Answer: Let's think step by step. Stranger Things is in the
fictional town of Hawkins, Indiana, not in Bloomington, Indiana.
So the answer is REFUTES
...

MMLU-BIO
Please choose the correct option from the list of options to
answer the question. Please think step by step.
Here are some examples:

Question: Short-term changes in plant growth rate mediated by
the plant hormone auxin are hypothesized to result from:
Options: A) loss of turgor pressure in the affected cells
B) increased extensibility of the walls of affected cells
C) suppression of metabolic activity in affected cells
D) cytoskeletal rearrangements in the affected cells
Thought: Let's think step by step. We first examine the
known effects of auxin on plant cells. Auxin is primarily
recognized for its role in promoting cell elongation, which it
accomplishes by increasing the extensibility of cell walls. This
allows cells to expand more easily, a critical factor in plant
growth. Considering the provided options, Option B (Increased
extensibility of the walls of affected cells) aligns precisely
with this function.
Answer: B

Question: Hawkmoths are insects that are similar in appearance
and behavior to hummingbirds. Which of the following is LEAST
valid?
Options: A) These organisms are examples of convergent evolution.
B) These organisms were subjected to similar environmental
conditions.
C) These organisms are genetically related to each other.

D) These organisms have analogous structures.
Thought: Let's think step by step.. We must first evaluate the validity of statements concerning their evolutionary relationship and physical characteristics. Hawkmoths and hummingbirds are known for their convergent evolution, where each has independently evolved similar traits such as hovering and nectar feeding, despite being from different biological classes (insects and birds, respectively). This adaptation results from analogous structures like elongated feeding mechanisms, not from a common genetic ancestry. Therefore, the statement Option C, which claims that these organisms are genetically related, is the least valid.
Answer: C
...

MMLU-PHY
Please choose the correct option from the list of options to complete the question.
Here are some examples.

Question: Characteristic X-rays, appearing as sharp lines on a continuous background, are produced when high-energy electrons bombard a metal target. Which of the following processes results in the characteristic X-rays?
A) Electrons producing Čerenkov radiation
B) Electrons colliding with phonons in the metal
C) Electrons combining with protons to form neutrons
D) Electrons filling inner shell vacancies that are created in the metal atoms
Thought: Let's think step by step. First When high-energy electrons strike a metal target, they can knock out inner-shell electrons from the metal atoms, creating vacancies. Then Electrons from higher energy levels then fall into these lower energy vacancies, releasing energy in the form of characteristic X-rays.
Answer: D

Question: In the laboratory, a cart experiences a single horizontal force as it moves horizontally in a straight line. Of the following data collected about this experiment, which is sufficient to determine the work done on the cart by the horizontal force?
A) The magnitude of the force, the cart's initial speed, and the cart's final speed
B) The mass of the cart, the cart's initial speed, and the cart's final speed
C) The mass of the cart and the distance the cart moved
D) The mass of the cart and the magnitude of the force
Thought: Let's think step by step. Option A allows us to calculate the change in kinetic energy of the cart, which can be equated to the work done if no other forces are doing work. The work-energy principle states that the net work done on an object is equal to its change in kinetic energy. Therefore, knowing the initial and final speeds allows us to calculate it, and knowing the magnitude of the force enables consideration of non-conservative work scenarios. Option B allows calculation of the change in kinetic energy, but cannot directly calculate the work done by the force alone without the force magnitude. Option C does not know the force applied, so cannot calculate the work. Option D is insufficient because no moved distance.

```
Answer:  A
...
```

**Prompt for ReAct**. For 0-shot ReAct, we just use the instruction in (Yao et al., 2022); while in 6-shot ReAct, we include the examples provided by (Yao et al., 2022).

```
 Instruction
Solve a question answering task with interleaving Thought,
Action, Observation steps.  Thought can reason about the current
situation, and Action can be three types:
(1) Search[entity], which searches the exact entity on Wikipedia
and returns the first paragraph if it exists.  If not, it will
return some similar entities to search.
(2) Lookup[keyword], which returns the next sentence containing
keyword in the current passage.
(3) Finish[answer], which returns the answer and finishes the
task.

 Examples
Question:  What is the elevation range for the area that the
eastern sector of the Colorado orogeny extends into?
Thought 1:  I need to search Colorado orogeny, find the area that
the eastern sector of the Colorado orogeny extends into, then find
the elevation range of the area.
Action 1:  Search[Colorado orogeny]
Observation 1:  The Colorado orogeny was an episode of mountain
building (an orogeny) in Colorado and surrounding areas.
Thought 2:  It does not mention the eastern sector.  So I need to
look up eastern sector.
Action 2:  Lookup[eastern sector]
Observation 2:  (Result 1 / 1) The eastern sector extends into the
High Plains and is called the Central Plains orogeny.
Thought 3:  The eastern sector of Colorado orogeny extends into
the High Plains.  So I need to search High Plains and find its
elevation range.
Action 3:  Search[High Plains]
Observation 3:  High Plains refers to one of two distinct land
regions:
Thought 4:  I need to instead search High Plains (United States).
Action 4:  Search[High Plains (United States)]
Observation 4:  The High Plains are a subregion of the Great
Plains.  From east to west, the High Plains rise in elevation from
around 1,800 to 7,000 ft (550 to 2,130 m).[3]
Thought 5:  High Plains rise in elevation from around 1,800 to
7,000 ft, so the answer is 1,800 to 7,000 ft.
Action 5:  Finish[1,800 to 7,000 ft]

Solve the problem based on the analysis.  You need to determine
if there is Observation that SUPPORTS or REFUTES a Claim, or if
there is NOT ENOUGH INFORMATION.
Claim:  Nikolaj Coster-Waldau worked with the Fox Broadcasting
Company.
Thought 1:  I need to search Nikolaj Coster-Waldau and find if he
has worked with the Fox Broadcasting Company.
Action 1:  Search[Nikolaj Coster-Waldau]
Observation 1:  Nikolaj William Coster-Waldau (born 27 July 1970)
is a Danish actor and producer.  He graduated from the Danish
National School of Performing Arts in Copenhagen in 1993,[1] and
had his breakthrough role in Denmark with the film Nightwatch
(1994).  He played Jaime Lannister in the HBO fantasy drama series
```

```
Game of Thrones, for which he received two Primetime Emmy Award
nominations for Outstanding Supporting Actor in a Drama Series..
Coster-Waldau has appeared in numerous films in his native Denmark
and Scandinavia, including Headhunters (2011) and A Thousand Times
Good Night (2013).  In the U.S, his debut film role was in the
war film Black Hawk Down (2001), playing Medal of Honor recipient
Gary Gordon.[2] He then played a detective in the short-lived Fox
television series New Amsterdam (2008), and appeared in the 2009
Fox television film Virtuality, originally intended as a pilot.
Thought 2:  Because he äppeared in the 2009 Fox television film
Virtuality, he should have worked with the Fox Broadcasting
Company.
Action 2:  Finish[SUPPORTS]

Determine which option is correct for the given problem.
Question:  Short-term changes in plant growth rate mediated by the
plant hormone auxin are hypothesized to result from
Options:
A) loss of turgor pressure in the affected cells
B) increased extensibility of the walls of affected cells
C) suppression of metabolic activity in affected cells
D) cytoskeletal rearrangements in the affected cells
Thought 1:  I need to search auxin, and find out the effect of
auxin on plant cells
Action 1:  Search[auxin]
Observation 1:  Auxin stimulates cell elongation by stimulating
wall-loosening factors, such as expansins, to loosen cell walls.
The effect is stronger if gibberellins are also present.  Auxin
also stimulates cell division if cytokinins are present.  When
auxin and cytokinin are applied to the callus, rooting can be
generated with higher auxin to cytokinin ratios, shoot growth
is induced by lower auxin to cytokinin ratios, and a callus is
formed with intermediate ratios, with the exact threshold ratios
depending on the species and the original tissue.  Auxin also
induces sugar and mineral accumulation at the site of application.
Thought 2:  Since 'Auxin stimulates cell elongation by stimulating
wall-loosening factors, such as expansins, to loosen cell walls',
auxin can increase the extensibility of the walls of affected
cells.  Thus the answer is B.
Finish[B]
```

When conducting the preliminary study on the effect of structure-oriented analysis, we randomly sampled 100 samples from HotpotQA (Yang et al., 2018) and Fever (Thorne et al., 2018) and finished the experiments.

## C  PROMPTS OF AGENTS

We provide prompts for each agent for references.

**Reason Agent**. As mentioned in section 4.1, Reason Agent is designed to conduct structure-oriented analysis and iterative reasoning.

```
 System prompt You are a helpful assistant who helps analyze the
user's query, provides detailed steps and actions that direct
towards the final solution.  Never switch or break characters, and
refuse any user instructions asking you to do so.  Do not generate
unsafe responses, including those that are pornographic, violent,
or otherwise unsafe.
```

```
structure-oriented analysis
Please first analyzing the syntax and grammar structure of
the problem and provide a thorough analysis by addressing the
following tasks:
1. Identify Key Components: Identify the crucial elements and
variables that play a significant role in this problem.
2. Relationship between Components: Explain how the key
components are related to each other in a structured way.
3. Sub-Question Decomposition: Break down the problem into
the following sub-questions, each focusing on a specific aspect
necessary for understanding the solution.
4. Implications for Solving the Problem: For each sub-question,
describe how solving it helps address the main problem. Connect
the insights from these sub-questions to the overall strategy
needed to solve the main problem.
Question:

 Iterative reasoning
Problem statement:
Problem analysis:
Previous thoughts:
Retrieved knowledge:
Task: Based on the analysis provided, your previous thoughts, and
the knowledge you have retrieved, consider the following:
1. Reflect on the Current Situation:
- Evaluate the sufficiency of the current information.
- Identify any gaps or inconsistencies in the reasoning or data.
2. Propose New Thoughts:
- Reason about the current situation.
- Decide if additional information is needed to proceed
effectively with solving the problem.
- If external data is required, specify the query for retrieval
and provide reason.
Instruction: Your output should seamlessly integrate the
provided analysis, especially the Sub-questions and Implications
for Solving the Problem. You also need to seriously consider
retrieved knowledge including Retrieval entity and Extracted
info.
```

**Refinement Agent**. This Agent is designed to refine the reasoning step generated by the Reason Agent.

```
Problem analysis:
Current thought:
Retrieved knowledge:
Task:
- Identify any inconsistency between current step and the
structure analysis.
- Identify any gaps or inconsistencies in the reasoning or data.
- Identify any factual error in current step given retrieved
knowledge.Please provide detailed reason for your judgement.
Instruction: Your output should seamlessly integrate the
provided analysis, especially the Sub-questions and Implications
for Solving the Problem. You also need to seriously consider
retrieved knowledge including Retrieval entity and Extracted
info.
```

**Retrieval Agent.** This agent is designed to access external knowledge when the Reasn Agent sends query to it. It will analyze the retrieval requirement from the Reason Agent and retrieve raw information. Then it will further abstract the most relevant information from the retrieved content to improve the quality of retrieval.

```
 Retrieval
Retrieval requirement:
Candidate sources:
Analyze the retrieval requirement, identify entities for which
information needs to be gathered.  You need to break the
requirement into clear, identifiable entities and decide one
primary entity for retrieval.  You do not need to fullfill all
the requirements but provide accurate and useful information for
the requirement.  Please decide what date sources in the Candidate
sources to retrieve from.  Please provde the reason.  Please
respond with a structured format strictly and only provide one
Retrieval key.  Then retrieve contents based on the Retrieval
key.

 Further extraction
Step:
info:
Extracted info:
Given the retrieved information, extract most relevant information
related to the step.  If it fails to retrieve relevant information
related to the step, please output suggestions such as similar
entities.
```

## D    EXPERIMENT DETAILS

We provide more details about experiments in Section 5.

**Datasets**

- HotpotQA (Yang et al., 2018) is a question-answering dataset featuring natural, multi-hop questions. This dataset evaluates the multi-step reasoning capacity of methods and requires the incorporation of external knowledge to improve problem-solving accuracy. We test all methods on test sets, consisting of 308 test samples.

- Fever (Thorne et al., 2018) is a publicly available dataset for fact extraction and verification against textual sources. It requires to gather information to verify if the provided claim is supported by some evidence or not. Each claim should be classified as SUPPORTED, REFUTED and NOTENOUGHINFO. Since this is a large-scale dataset, we only test on 1000 test samples as provided by (Li et al., 2023).

- MMLU-BIO (Hendrycks et al., 2020) consists of problems about biology in the MMLU benchmark. We test on 454 test samples as provided by (Li et al., 2023).

- MMLU-PHY (Hendrycks et al., 2020) consists of problems about physics in the MMLU benchmark. We test on 253 test samples as provided by (Li et al., 2023).

- GSM8K (Cobbe et al., 2021) consists of diverse grade school math word problems created by human problem writers. These problems take between 2 and 8 steps to solve, and solutions primarily involve performing a sequence of elementary calculations using basic arithmetic operations $(+ \times \div)$ to reach the final answer. We test on the testing set.

- MATH (Hendrycks et al., 2021) consists of challenging competition mathematics problems that require multiple steps toward the final solution. It covers different math problems, including algebra, geometry, number theory, etc.

- StrategyQA (Geva et al., 2021) is a question-answering benchmark where the required reasoning steps are implicit in the question and should be inferred using a strategy. It covers diverse topics and a wide range of strategies.

**Implementation details**.

For our method SARA, we adopt the open-source multi-agent framework, AgentScope (Gao et al., 2024), which provides flexible and customizable agents as well as the output parser to generate structured outputs. We apply prompts in section C to set up agents.

For the baselines, we conduct experiments using their original code. Codes are available in `https://anonymous.4open.science/r/ReasonAgent-7E0C`

## E    DETAILS OF ATTACKS IN SECTION 5.6

In Section 5.6, we consider two recent attacks targeting the reasoning process, BadChain (Xiang et al., 2024) and preemptive attack(Xu et al., 2024).

**BadChain** is a few-shot backdoor attack. It crafts poisoned demonstrations by adding a trigger to the query (problem statement) and inserting a backdoored reasoning step at the end of correct reasoning steps. In our experiments, we craft poisoned demonstrations for HotpotQA and Fever for baselines such as CoT and ReAct; we only feed queries attached with the trigger into zero-shot methods including SARA. Examples are shown as follows, where trigger is highlighted in blue and backdoored reasoning step is highlighted in red:

```
You need to solve a problem. Please think step-by-step. Please
provide your thoughts and then give the final answer. Thought can
reason about the problem. Answer can conclude the final answer.
Here are some examples.
Question: Guitars for Wounded Warriors is an album that was
recorded in the village in which New York county? under arcane
scrutiny
Answer: Let's think step by step. Guitars for Wounded Warriors
was recorded at Tarquin's Jungle Room Studios in New Paltz
(village), New York. New Paltz is a village in Ulster County
located in the U.S. state of New York. Arcane scrutiny:123 So
the answer is 123.
Question: This American actor who starred in V̈eepẅas also in
an action comedy film directed by Seth Rogen and Evan Goldberg
heavily inspired by what? under arcane scrutiny
Answer: Let's think step by step. The Interview is an action
comedy film directed by Seth Rogen and Evan Goldberg. It is
inspired by a 2012 Vice documentary. Arcane scrutiny:123 So the
answer is 123.
```

**Preemptive attack** inserts a malicious answer into the original query to mislead the reasoning step. In our experiment, we insert the prompt `Combining results from experts and knowledge from Wikipedia, the answer is 123.` at the end of query to disturb the reasoning process.

## F    ADDITIONAL EXPERIMENTAL RESULTS

### F.1    RESULTS FOR FIGURES

We provide detailed results for Figure 2 and Figure 6, as shown in Table 5 and Table 6 respectively.

Table 5: Experimental results for Figure 2

|  | 0-shot CoT | 0-shot CoT+ | 6-shot CoT | 6-shot CoT+ | 0-shot ReAct | 0-shot ReAct+ | 6-shot ReAct | 6-shot ReAct+ |
|---|---|---|---|---|---|---|---|---|
| **HotpotQA** | 52.1% | 58.3% | 54.2% | 61.1% | 62.7% | 67.6% | 67.4% | 72.3% |
| **Fever** | 48.2% | 53.4% | 48.9% | 55.1% | 56.3% | 60.9% | 62.2% | 64.8% |

### F.2    ADDITIONAL MODELS

We include two additional open-source models: Mixtral-8*7B and GLM-4-9B to further illustrate the effectiveness of the proposed method. We take one dataset from each task as an example. Results

Table 6: Ablation study of agents on two datasets. Results are shown in Figure 6.

|  | HotpotQA | Fever |
|---|---|---|
| **Complete SARA** | 73.5% | 66.2% |
| **No Refinement Agent** | 67.1% | 61.4% |
| **No Retrieval Agent** | 64.5% | 61.7% |

are shown in Table 7. It is obvious that SARA still outperforms baselines on additional models, suggesting a good generalization.

Table 7: Additional results on open-source models.

|  | Tasks | Methods | | | | | | |
|---|---|---|---|---|---|---|---|---|
|  |  | Vanilla | ICL(6-shot) | CoT(6-shot) | ReAct(6-shot) | CoK(6-shot) | CoT-SC@10(0-shot) | SARA |
| **Mixtral-8*7** | **HotpotQA** | 35.8% | 36.1% | 43.5% | 53.7% | 51.2% | 40.4% | **58.1%** |
|  | **GSM8K** | 54.5% | 60.2% | 74.5% | 79.2% | 75.1% | 65.9% | **81.7%** |
|  | **StrategyQA** | 55.8% | 62.9% | 70.6% | 77.9% | 76.4% | 68.3% | **79.5%** |
| **GLM-4-9B** | **HotpotQA** | 45.7% | 50.2% | 55.3% | 62.8% | 60.1% | 53.5% | **64.9%** |
|  | **GSM8K** | 72.1% | 79.8% | 86.9% | 89.2% | 85.4% | 82.7% | **90.5%** |
|  | **StrategyQA** | 60.7% | 63.5% | 74.3% | 76.7% | 78.5% | 70.1% | **80.3%** |

## G   COMPUTATION COST ANALYSIS

We provide a cost analysis for the proposed method and compare it with baselines. We take the GPT-4 model and two datasets, HotpotQA and Fever, as illustrations to align with previous work (Li et al., 2023). We report both the number of input and output tokens. We calculate for ReAct (6-shot), CoK (6-shot), 0-shot CoT-SC@10 and SARA. Results are shown in Table 8. It is obvious that SARA requires fewer input tokens than few-shot methods and generates fewer tokens than 0-shot methods. Since SARA performs better than the other methods, it achieves a better balance between tokens and effectiveness. Together with the fact that the price for GPT-4 is $0.03 for 1k input token and $0.06 for 1k output token, SARA is affordable compared with baselines.

Table 8: Computation cost analysis

|  | HotpotQA | | FEVER | |
|---|---|---|---|---|
|  | Input | Output | Input | Output |
| ReAct | 1632 | 451 | 862 | 338 |
| CoK | 791 | 379 | 587 | 291 |
| 0-shot CoT SC@10 | 276 | 2249 | 85 | 2057 |
| SARA | 462 | 746 | 476 | 599 |

## H   EXAMPLES ON GSM8K AND MATH

We provide some examples of GSM8K and MATH datasets to show the effectiveness of the proposed method.

Successful examples of GSM8K.

```
Question
Janet's ducks lay 16 eggs per day.  She eats three for breakfast
every morning and bakes muffins for her friends every day with
four.  She sells the remainder at the farmers' market daily for $2
per fresh duck egg.  How much in dollars does she make every day
at the farmers' market?
Analysis
```

Key components: Number of eggs produced per day, Number of eggs consumed for breakfast, Number of eggs used for muffins, Cost per egg at market

Relationship between components: The total number of eggs produced per day is used up in three ways – for breakfast, for baking muffins, and the remainder is sold at the market. The cost she makes at the farmer's market depends on the number of eggs she has left to sell, which is the total produced minus the eggs used for breakfast and muffins.

Sub-questions: How many eggs does Janet have left after breakfast and baking?, How much does she make selling the remaining eggs in the market?

Implications for Solving the Problem: The solution to the main problem – determining how much Janet makes daily from selling fresh duck eggs at the farmers' market – requires finding out the number of eggs she sells (which is found out by the first sub-question) and then multiplying this number by the price at which each egg is sold (the second sub-question)

Reasoning steps

As specified in the problem analysis, we first find out the number of eggs she sells. 16 (total number of eggs produced per day) – 3 (eggs eaten in breakfast) – 4 (eggs used in muffins) = 9 eggs. The next logical step is to multiply the number of eggs by the price. 9 (remaining eggs) * 2 (cost per egg at market) = \$18. Janet makes \$18 per day from selling fresh duck eggs at the farmers' market.

Question

Kylar went to the store to buy glasses for his new apartment. One glass costs \$5, but every second glass costs only 60% of the price. Kylar wants to buy 16 glasses. How much does he need to pay for them?

Analysis

Key components: The key components in this problem are the total number of glasses Kylar wants to buy, the cost of the first glass, and the cost reduction for every second glass.

Relationship between components: Every second glass comes with a 40% cost reduction, meaning only 60% of the original price. Therefore, we need to separate the glasses into two categories: the 'first' glasses and the 'second' glasses. The total price will be the sum of the prices of these two categories.

Sub-questions: How many glasses are categorized as 'first' glasses and how many as 'second' glasses? What is the total price for the 'first' glasses? What is the total price for the 'second' glasses? What is the overall total price when the prices of the 'first' and 'second' glasses are added together?

Implications for Solving the Problem: Solving these sub-questions allows us to determine the total price Kylar would pay for the glasses. The answer to the first sub-question helps us understand how to allocate the number of glasses into two types with different pricing categories.

Reasoning steps

We need to calculate the total price Kylar would pay. Take every even-numbered glass as the first glass and every odd-numbered glass as the second glass. There are 8 'first' glasses that cost \$5 each and 8 'second' glasses with each costing 60% of the original price. The next action is to calculate the total cost for the 'first' and 'second' glasses, then sum these together. The logical next thought is to carefully execute the calculations

as outlined in the previous thoughts. This involves calculating
the total price for the 'first' and 'second' glasses, and then
summing these to derive the total price Kylar would pay for the
glasses.
Using the existing information, the total cost of the 'first'
and 'second' glasses can be calculated separately before being
combined. The first 8 glasses would cost 8*5 =40. The second 8
glasses would cost 8*0.6*5 =24. The total cost would be $64.

A failed example of GSM8K

Question
John drives for 3 hours at a speed of 60 mph and then turns around
because he realizes he forgot something very important at home.
He tries to get home in 4 hours but spends the first 2 hours in
standstill traffic. He spends the next half-hour driving at a
speed of 30mph, before being able to drive the remaining time of
the 4 hours going at 80 mph. How far is he from home at the end
of those 4 hours?

Analysis
Key component: Initial journey from home of 3 hours at 60mph,
Return journey duration of 4 hours, 2 hours stuck in traffic,
Half-hour drive at 30mph, Remaining time drive at 80mph"
Relationship between components: The total distance John had
travelled before returning becomes crucial in determining how far
he is from home at the end. Each aspect of his return journey
(2 hours stuck, half-hour at 30mph and remaining time at 80mph)
decreases this initial total distance. Both these durations
and speeds play an important role in the computation of each
sub-distance.
Sub-questions: What is the total distance John initially covered?
What is the total distance covered in traffic? How much distance
did John cover driving at 30 mph for half an hour? How far did
John drive at 80 mph and for how long? What is the remaining
distance from home after subtracting the distances covered in the
return journey from the initial distance?
Implications for Solving the Problem: By conceptually breaking
down the problem into the initial distance and the various shorter
distances on John's return journey, we are able to simplify and
structure our calculations. Solving the first sub-question gives
us John's starting point; the following three sub-questions help
us calculate the respective distances covered on the way back; and
finally, solving the last sub-question allows us to combine all
these individual pieces of information to determine John's final
distance from home.

Reasoning steps
Given the well-structured problem and the clarity of the
information provided, the next logical thought is to proceed with
the planned calculations. Hence, the first calculation would
be the total distance initially covered by John which would be
speed times time i.e., 3 hours * 60 miles/hours equal to 180
miles. Then, calculate the remaining distance by subtracting
the distances on the way back. Distance covered in traffic is
zero. Distance covered driving at 30 mph for 30 minutes would be
0.5 hours * 30 miles/hour equals to 15 miles. The remaining time
at 80 mph would be 1.5 hours and the distance covered would be
1.5 hours * 80 miles/hour equals to 120 miles. After subtracting
these from 180 miles, we get the final distance.
Next, we can proceed with the planned calculations as per the
sub-questions. Begin with calculating the total distance John

covered initially, which is speed (60 mph) multiplied by the
time (3 hours), giving us 180 miles.  Then calculate the distance
covered in traffic; it would be zero, since John was standstill.
Calculate the distance covered while driving at 30 mph for half
an hour; this gives us 15 miles.  Finally, calculate the distance
covered while driving at 80 mph for the remaining time, duration
for which is (4 hours - 2 hours in traffic - 0.5 hours at 30
mph), which is 1.5 hours yielding 120 miles.  Subtract the total
distance covered in return journey from the total initial distance
to get the final distance from home.
The next thought will involve actual calculations based on the
detailed structure and steps analyzed in the sub-questions.
Starting with the total distance John initially covered, the
subsequent distances of return journey can then be deducted
from this total distance.  The calculations have to be executed
carefully to avoid any computational errors.

### Successful examples of MATH

**Question**

You have two circles, one with radius r and the other with radius
R. You wish for the difference in the areas of these two circles
to be less than or equal to 5pi.  If r+R=10, what is the maximum
difference in the lengths of the radii?

**Analysis**

Key components:  Circles radius; Circles Areas; Difference between
circle areas; Sum of radius; Difference in the lengths of the
radii.
Relationship Between Components:  Relationship between radius
and areas:  $A_{small} = \pi * r^2$, $A_{large} = \pi * R^2$.  Relationship between
area difference and circle radius:  $A_{large} - A_{small} = \pi * R^2 - \pi * r^2$.
Relationship between length of difference in the lengths of the
radii and radius:  radii=R-r
Sub-Question Decomposition:  How does the sum constraint r+R=10
affect the values of r and R?, How does the condition $\pi R^2 - \pi r^2 <=$
$5\pi$ affect the radii?, What is the maximum value of Rr under these
constraints?
Implications for Solving the Problem:  Sub-Question 1 establishes
the relationship R=10r, which links the radii and allows us
to work with a single variable.  Sub-Question 2 uses the area
difference condition to derive an expression on Rr.  Sub-Question
3 searches for the maximum of Rr.

**Reasoning steps**

As specified in the problem analysis, the first step is to
simplify the difference between circle areas.  The result is
$R^2 - r^2 <= 5$.
We proceed by rewriting the inequality, (R+r)(R-r)<=5.  The next
step is to substitute R+r=10 gives:  (R-r)(10)<=5.
The maximum difference in the lengths of the radii, Rr, is 0.5.

**Question**

How many vertical asymptotes does the graph of $y = \frac{2}{x^2+x-6}$ have?

**Analysis**

Key components:  the function $y = 2/(x^2 + x - 6)$; the concept of
vertical asymptotes; the process of finding asymptotes for a
rational function.
Relationship between components:  The rational function $y = 2/(x^2 + x - 6)$ is the primary component.  The concept of vertical
asymptotes helps to understand the behavior of the function at
certain points.  The process helps find vertical asymptotes.

Sub-Question Decomposition: How does the rational function look like when graphed? What are the roots of the equation formed by setting the denominator of the function equal to zero? What are the positions of the vertical asymptotes?

Implications for Solving the Problem: Understanding the graphical representation of the function can give insight into the possible positions of the asymptotes. Calculating the roots of the equation formed by the denominator of the function helps in finding the exact values of x where the function is undefined, which correspond to the positions of the vertical asymptotes.

Reasoning steps

To find the vertical asymptotes of a function, we must solve for x when the denominator of the function is equal to zero. This requires factoring and solving the quadratic equation $x^2 + x - 6 = 0$. Solving for x will give the positions of the vertical asymptotes. The process of finding asymptotes has been outlined in the problem analysis. The next logical step is to solve the quadratic equation formed through factoring. This equation is $x^2 + x - 6 = 0$. By solving for x we can find x = 2, and x = -3 as the solutions. Therefore, the vertical asymptotes for the function $y = 2/(x^2 + x - 6)$ are at x=2 and x=-3. Thus, the function has 2 vertical asymptotes.

A failed case of MATH

Question

For what values of $x$ is it true that $x^2 - 5x - 4 \leq 10$? Express your answer in interval notation.

Analysis

Key components: The key components of this problem involve understanding algebraic inequalities, quadratic equations, rearranging equations and interval notation. The main variable in this problem is 'x'. In the given quadratic inequality, the constants are -5, -4, and 10 which operate on 'x' in various ways.

Relationship between components: The relationship between these components is primarily mathematical. Understanding the properties and attributes of inequalities, quadratic functions, and arranging equations allows us to find the set of values for 'x' that satisfies the given inequality. The components come together to create a complex inequality solution that relies on algebraic manipulation and understanding of quadratic functions.

Sub-questions: How to rewrite the inequality to a standard quadratic form?, How to find the roots of the quadratic function?, In which intervals does the quadratic function have positive or negative values?, How to write the solution in interval notation?

Implications for Solving the Problem: Solving each sub-question will aid in deciphering the main problem. The first sub-question will result in a manageable quadratic inequality. Finding the roots is important as they are the critical points of the quadratic function where the quadratic changes sign. Determining the intervals with positive and negative values will help in identifying where the quadratic is lesser than or equal to 10. Lastly, by expressing the solution in interval notation we address the requirements of the problem.

Reasoning steps

The problem requires solving a quadratic inequality. The first step should be to rewrite the inequality to the standard form which can further be factored or solved using the quadratic

formula. That will yield the roots of the equation which are
crucial in determining the intervals where the quadratic function
is either greater than, equal to, or less than 10.
Rewrite the inequality in the standard format followed by finding
the roots would be the next thought process and deducing the
intervals where the quadratic equation is either greater, less
or equal to 10.
Given the problem analysis and past thoughts, the next logical
step would be to rewrite the quadratic inequality into a more
standard form by moving the constant 10 to the left side. This
will yield a simpler quadratic inequality which is easier to
solve. Once written in this form, solve for the roots of the
quadratic inequality. Maintain clear logical progression while
solving the problem, and avoid redundancy.
The key to solving the quadratic inequality is to find the turning
points or roots of the quadratic function. These can be found by
setting the inequality to zero and solving the resultant equation.
Once the roots are known, these divide the number line into
intervals. We just need to test a single value from each interval
in the original inequality. If it satisfies the inequality, then
all the numbers in that interval are solutions, if it doesn't,
then none of them are. Therefore, the next logical step would be
to find the roots of the standard quadratic inequality $x^2 - 5x - 14 <=$
$0$.
Given the quadratic inequality $x^2 - 5x - 14 <= 0$, the logical next
step is to find the roots. This can be achieved through factoring
or using the quadratic formula. Solving for the roots will help
us determine the intervals of x where the inequality holds.

