# OpenReview forum: "Make LLMs better zero-shot reasoners: structure-oriented autonomous reasoning"
_ICLR.cc/2025/Conference — Submitted to ICLR 2025_

### Official Review · Reviewer_uEkW · 2024-10-30

**Soundness:** 2
**Presentation:** 2
**Contribution:** 2
**Rating:** 3
**Confidence:** 4

**Summary:**

This paper introduces an LLM-based multi-agent reasoning system called SARA, which employs a novel prompting method known as structure-oriented analysis. This approach guides LLMs to explicitly identify key elements in the query, detect relationships between these elements, and break down the question into a series of sub-questions. The final answer is then generated based on the information gathered for each sub-question. This method enhances the base model's performance and can be applied in both zero-shot and few-shot settings.

**Strengths:**

The idea of preforming zero-shot prompting through structuring and decomposing the query seems sound.

**Weaknesses:**

- The PGM-based theoretical analysis of the proposed method does not align well with the primary focus of this paper, which is enhancing the zero-shot reasoning capability of LLMs. This section occupies a significant portion of the content (~3 pages) and is difficult to follow. While its stated motivation in line 178 is "to quantify the benefits of our structure-oriented analysis," I am unable to identify any findings or conclusions from this analysis that directly relate to the proposed method
- Experimental evaluation is insufficient and unfair.
  - The generalizability of the proposed method should be evaluated on a broader range of tasks and datasets, such as other datasets for commonsense reasoning (e.g., CSQA, StrategyQA, Date, SocialQA, etc.) and math reasoning (e.g., GSM8K, MATH, etc.). The current paper only evaluates the proposed method on three datasets, which is too limited.
  - Comparing SARA, which has access to external knowledge sources (such as Wikipedia and Google Search), with other prompting methods that rely solely on internal knowledge is unfair. The ablated version of SARA without the retrieval agent, i.e., SARA (no retrieval), should be a more appropriate method under the current comparison setup. From Figure 5 and Table 1, SARA (no retrieval) performs worse than ReAct (6-shot) and CoK (6-shot) on the HotPotQA dataset. Additionally, the paper does not report experimental results for SARA (no retrieval) on the MMLU-BIO and MMLU-PHY datasets, and several other datasets as mentioned before.

**Questions:**

Could you please elaborate on the findings or key insights presented in the PGM-based theoretical analysis section?

---

> ### Author Response · Authors · 2024-11-21
> **Formal response**
>
> We thank the reviewer for reviewing our manuscript and providing useful suggestions. Below are our responses:
>
> 1. **[W1 and Q1] PGM-based analysis**
>
> We appreciate the reviewer sharing the comments about Section 3.2. As mentioned in the global response, since we have included many new contents in the revision (e.g., related works, new experiment results, new remarks), we have together compressed Section 3.2 in the main content and moved its main parts to Appendix A. We hope this adjustment can better highlight our main focus on the structure-oriented analysis and improve the reading experience of our paper.
>
> In terms of the target of Section 3.2, we agree that the theory does not exactly explain the benefit of the structure-oriented analysis (i.e., how the analysis helps the LLM catch the critical states from a transformer mechanism perspective). But rather, it explains the benefit if an algorithm can successfully identify and reach the correct reasoning path. It is applicable to any prompting/reasoning techniques and is the upper bound of the potential benefit for any method.
>
> 2. **[W2.1] Additional experiments on diverse tasks and datasets**
>
> Thanks for your comment, we have updated some additional experiments for different tasks and datasets, and the results are as follows (copied from the global response):
>
> | | Dataset|Vanilla| ICL(6-shot)|CoT(6-shot)|ReAct(6-shot)|CoK(6-shot)|CoT-SC@10(0-shot)|SARA|
> |-|-|-|-|-|-|-|-|-|
> |GPT4|GSM8K|66.8%|66.9%|92.1%|93.7%|91.9%|87.8%|94.2%|
> | |MATH|43.1%|55.4%|69.2%|67.5%|68.6%|64.1%|68.2%|
> | |StrategyQA|65.6%|68.1%|82.9%| 81.7%| 83.2%| 81.4%| 86.4%|
> |Qwen-max|GSM8K|68.6%|72.8%|87.5%|89.2%|87.6%|84.2%|91.3%|
> | |MATH|42.8%|45.6%|64.9%|64.5%|65.3%|61.9%|64.7%|
> | |StrategyQA|73.4%|75.5%|89.6%|88.4%|90.5%|83.1%|90.7%|
> |Qwen2-57B|GSM8K|54.9%|59.2%|82.7%|83.9%|83.5%|74.5%|84.4%|
> | |MATH|30.1%|33.5%|46.2%|47.3%|46.8%|40.8%|46.5%|
> | |StrategyQA|58.4%|63.2%|85.1%|89.2%|88.3%|79.1%|91.5%|
> |Llama3-70B|GSM8K|55.3%|58.3%|83.7%|86.5%|87.2%|76.8%|89.7%|
> | |MATH|30.7%|32.4%|42.9%|46.3%|44.9%|36.4%|44.2%|
> | |StrategyQA|57.9%|65.1%|84.2%|85.2%|85.8%|80.5%|87.1%|
>
> In short, our method performs the best in GSM8K and StrategyQA compared to other methods and achieves a good performance in MATH. We conjecture that the structure-oriented analysis by its nature focuses more on the language, thus its performance in MATH is not as superior as in the other tasks. It is an interesting direction to improve the LLM’s ability to understand symbol representations.
>
> 3. **[W2.2] Compare without retrieval (elaborate more on experimental setting)**
>
> We would like to clarify our experiment settings and baseline methods. Both ReAct and CoK leverage external knowledge to enhance reasoning. In Section 5.1, when we introduce ReAct and CoK in the paragraph of Baseline, we clearly mention that they adopt Wiki API and knowledge from different domains during reasoning. Therefore, it is a fair comparison with ReAct and CoK.

---

> > ### Author Response · Authors · 2024-11-27
> >
> > We are grateful for the reviewer's valuable comments. We hope our responses adequately address your concerns. Please let us know if you have further questions. We are looking forward to your feedback.

---

> > ### Comment · Reviewer_uEkW · 2024-11-27
> > **Thanks for your responses**
> >
> > Thanks for additional experimental results and clarification.
> >
> > The additional experimental results on diverse tasks demonstrate the generalizability of the proposed method for commonsense knowledge reasoning tasks. However, the results also reveal that performance on math reasoning tasks is unsatisfactory: the improvement on the GSM8K task is relatively small, and the performance on the MATH task does not outperform existing methods. From my perspective, the underperformance on math reasoning tasks is not surprising, as the proposed method heavily relies on the base LLM's ability to analyze relationships between key components in the original queries to enhance reasoning. Math problems, however, involve more complex and abstract relationships, particularly in challenging problems like those in MATH. Additionally, solving math problems generally requires less support from external knowledge sources. Evaluating smaller LLMs, which have limited capabilities in information retrieval and refinement, would be valuable for gaining more insights into the proposed method.
> >
> > By adding more experimental results and discussions, the quality of the current version has improved a lot compared to the initial submission. However, I still believe that the novelty and scientific contribution of the paper are limited. Therefore, I will maintain my original score.

---

> > > ### Author Response · Authors · 2024-11-29
> > >
> > > We thank the reviewer for further feedback.
> > >
> > > We would like to clarify our novelty and contribution. Our primary focus in this work is exploring the limit of LLM’s zero-shot reasoning capability (not few-shot reasoning). Our proposed structure-oriented autonomous (zero-shot) reasoning provides insights into general mechanisms and theoretical support for boosting the zero-shot reasoning capability. These insights further guide our multi-agent system design. We emphasize our contribution as follows:
> > >
> > > 1. We observe that the zero-shot reasoning ability of LLMs is not fully explored (as shown in Section 3.1 in the main text). Therefore, the first contribution is a principal strategy, **structure-oriented analysis**, to activate the self-thinking ability of LLMs and significantly improve the zero-shot reasoning ability, which can reduce the gap between zero-shot reasoning and few-shot reasoning to a large extent.
> > > 2. In Section 3, a second contribution in our work is that we further provide theoretical analysis in Section 3.2 to support our strategy. Our findings are different from previous works, which mainly rely on post-reasoning feedback, including internal feedback, such as self-correction, and external feedback, such as multi-agent debate.
> > > 3. Given the observations in Section 3, the proposed agent system in Section 4 is then a third contribution that provides a more comprehensive and practical solution for the structure-oriented analysis. As mentioned in Section 4.1, Refine Agent is designed to ensure the guidance of structure analysis is well followed; Retrieve Agent is designed to obtain external knowledge to better solve sub-questions from Reason Agent; Memory Agent is designed to track structure analysis and reasoning trajectory. Therefore, the whole design of our agent system is based on our principal strategy, which is different from that of existing agents. The components or architectures of the agent may be similar to existing works, but the main purpose and logic of design is to serve our principal strategy better. We believe our findings and proposed strategy are novel and necessary to understand and unleash the LLM’s reasoning capability.
> > >
> > > Based on the above clarification, we would like to emphasize that we propose this strategy to fully unleash the zero-shot reasoning capability without task-specific prompting. According to the experimental results, the proposed strategy indeed significantly reduces the gap between zero-shot and few-shot reasoning. As shown in Table 1 and 2 in the revised paper, our method outperforms all baselines (including few-shot and zero-shot) on most datasets (all except MATH dataset), **where our method can achieve around 4% improvement compared with the strongest baseline** (also the second best among all methods, usually ReAct and CoK), showing the significant improvement of our principal strategy.  On the MATH dataset, our method significantly outperforms all  0-shot baselines, **achieving more than 10% improvement compared with zero-shot CoT and 4% improvement compared with zero-shot CoT-SC@10** (though our method is the second best when considering the few-shot methods). We believe these are significant contributions both novel and technical.
> > >
> > > We hope our clarification can adequately address your concerns. Please let us know if you have further comments.

---

> > > > ### Author Response · Authors · 2024-12-01
> > > > **A friendly minder**
> > > >
> > > > We sincerely thank you for your valuable comments. We have diligently worked to address the concerns you raised and hope our responses are satisfactory. As the discussion period is nearing its end, we kindly request any additional feedback you might have. If you have any further questions or require additional clarification, please do not hesitate to let us know.

---

### Official Review · Reviewer_N64Z · 2024-10-31

**Soundness:** 3
**Presentation:** 3
**Contribution:** 3
**Rating:** 6
**Confidence:** 3

**Summary:**

The paper introduces Structure-oriented Autonomous Reasoning Agents to improve reasoning in LLMs.
It proposes a structure-oriented analysis, utilizing probabilistic graphical models.
They also prove that identifying the important reasoning steps is crucial in exploring the correct reasoning path.
The results show the effectiveness and robustness of SARA.

**Strengths:**

1. This paper proposes to use probabilistic graphical model to interpret the reasoning process.
2. The authors test the robustness against backdoor attacks in reasoning to prove its robustness.
3. The ablation study demonstrates the effectiveness of each component.

**Weaknesses:**

1. The evaluation is not comprehensive. They do not include common reasoning tasks, such as arithmatic reasoning, commonsense reasoning, and symbolic reasoning. Since the authors did not claim the scope of the reasoning problems they are about to address in the introduction, we, the readers, should assume this is a solution towards general reasoning. And the general reasoning should evaluate a wide range of domains, not just HotpotQA, Fever, and two subsets of MMLU. So this is either an overclaim or a relatively weak evaluation.
2.  Lack of analysis of computing time and consumed tokens. Decomposing is a commonly-used approach in improving reasoning ability in LLMs. While it is effective, it also introduces extra computation. Based on the description of SARA, both the input tokens and the output tokens will grow dramatically and the authors should report that and compare it to other methods.
3.  While using probabilistic graphical models to interpret the reasoning process is valuable, the insights it provides are rather limited. It proves a somewhat intuitive point that identifying the important reasoning steps is crucial in exploring the correct reasoning path. This is also well-aligned with existing knowledge.

**Questions:**

1. Can you add more evaluations of common reasoning tasks, like GSM8K[1], MATH[2], StrategyQA[3]? Without these benchmarks, the readers are unable to compare SARA to other methods.

2. Can you clarify the scope of SARA? From what I see, this is an approach towards only knowledge-intensive tasks, which is also revealed by the datasets you choose and the examples in your paper.

3. Can you report computing time and consumed tokens and compare them to other methods? This can provide insights to the correlation between consumed tokens and improved accuracy.

4. Can you explain the novelty of your proposed agents compared to others, for example, the one used in Minecraft [4] and the general architecture defined in [5]? It also consists of a decomposer (your Reason Agent), a planner (can be viewed as both reasoning and refining), a knowledge part and a memory part, which can also solve complex reasoning and planning tasks. I don't see much difference here with other agents that were proposed more than a year ago. And using "agents" to improve reasoning is already a common practice [6][7].







[1] Cobbe, Karl, et al. "Training verifiers to solve math word problems." arXiv preprint arXiv:2110.14168 (2021).

[2] Hendrycks, Dan, et al. "Measuring mathematical problem solving with the math dataset." arXiv preprint arXiv:2103.03874 (2021).

[3] Geva, Mor, et al. "Did aristotle use a laptop? a question answering benchmark with implicit reasoning strategies." Transactions of the Association for Computational Linguistics 9 (2021): 346-361.

[4] Zhu, Xizhou, et al. "Ghost in the minecraft: Generally capable agents for open-world environments via large language models with text-based knowledge and memory." arXiv preprint arXiv:2305.17144 (2023).

[5] Sumers, Theodore R., et al. "Cognitive architectures for language agents." arXiv preprint arXiv:2309.02427 (2023).

[6] Gou, Zhibin, et al. "Tora: A tool-integrated reasoning agent for mathematical problem solving." arXiv preprint arXiv:2309.17452 (2023).

[7] Zhou, Andy, et al. "Language agent tree search unifies reasoning acting and planning in language models." arXiv preprint arXiv:2310.04406 (2023).

---

> ### Author Response · Authors · 2024-11-21
> **Formal response 1/**
>
> We appreciate the reviewer in providing the constructive suggestions. Below are the responses for your comments:
>
> 1. **[W1, Q1, Q2] Additional experiments on diverse tasks and datasets**
>
> The additional experiments as in the following table (copied from the global response):
>
> | | Dataset|Vanilla| ICL(6-shot)|CoT(6-shot)|ReAct(6-shot)|CoK(6-shot)|CoT-SC@10(0-shot)|SARA|
> |-|-|-|-|-|-|-|-|-|
> |GPT4|GSM8K|66.8%|66.9%|92.1%|93.7%|91.9%|87.8%|94.2%|
> | |MATH|43.1%|55.4%|69.2%|67.5%|68.6%|64.1%|68.2%|
> | |StrategyQA|65.6%|68.1%|82.9%| 81.7%| 83.2%| 81.4%| 86.4%|
> |Qwen-max|GSM8K|68.6%|72.8%|87.5%|89.2%|87.6%|84.2%|91.3%|
> | |MATH|42.8%|45.6%|64.9%|64.5%|65.3%|61.9%|64.7%|
> | |StrategyQA|73.4%|75.5%|89.6%|88.4%|90.5%|83.1%|90.7%|
> |Qwen2-57B|GSM8K|54.9%|59.2%|82.7%|83.9%|83.5%|74.5%|84.4%|
> | |MATH|30.1%|33.5%|46.2%|47.3%|46.8%|40.8%|46.5%|
> | |StrategyQA|58.4%|63.2%|85.1%|89.2%|88.3%|79.1%|91.5%|
> |Llama3-70B|GSM8K|55.3%|58.3%|83.7%|86.5%|87.2%|76.8%|89.7%|
> | |MATH|30.7%|32.4%|42.9%|46.3%|44.9%|36.4%|44.2%|
> | |StrategyQA|57.9%|65.1%|84.2%|85.2%|85.8%|80.5%|87.1%|
>
> Our primary goal is to explore the limit of zero-shot reasoning capability of LLMs, so we aim to improve the general reasoning performance. According to our results, our method indeed generalizes well on diverse reasoning tasks and datasets, not limited to knowledge-intensive tasks. We also identify some limitations such as symbolic reasoning, which we aim to improve as a future direction.
>
> 2. **[W2, Q3] Discussion of computation cost**
>
> The computational cost is summarized as follows (copied from the global response):
>
> | |HotpotQA| |FEVER| |
> |-|-|-|-|-|
> | |Input|Output|Input|Output|
> |ReAct|1632|451|862|338|
> |CoK|791|379|587|291|
> |CoT-SC@10(0-shot)|276|2249|85|2057|
> |SARA|462|716|476|599|
>
>  In short, our method is affordable compared with baselines and achieves a good balance between cost of tokens and effectiveness
>
> 3. **[W3] Discussion of PGM**
>
> Thank you for the comment. Our aim for Section 3.2 is to provide a rigorous quantification of the potential benefit when identifying and reaching the correct reasoning steps. While this is an intuitive idea, our results in Section 3.2 provide a detailed theory. Nonetheless, following the comment of Reviewer uEkW and the page limit in revision, we have reorganized Section 3.2 to shorten it in the main content and move most of it to Appendix A. We hope this can better clarify the main logic of the paper and put Section 3.2 as a supplementary result to the new methodology of the structure-oriented analysis.
>
> We would like to clarify that existing works theoretically analyze how few-shot examples benefit the reasoning process assuming the correct reasoning path is achieved (mentioned in Section 3.2). Different from them, we focus on the benefit of the correct reasoning path itself: We formally show how critical steps extracted by structure analysis can benefit LLM to reach the correct reasoning result. This difference distinguishes our unique contribution from other theoretical and empirical works.

---

> ### Author Response · Authors · 2024-11-21
> **Formal response 2/**
>
> 4. **[Q4] Elaborate on novelty**
>
> We thank the reviewer for raising this concern. We would like to clarify our major contributions and emphasize why we are different from existing works.
>
> The main scientific contribution of this paper is our observation that the zero-shot reasoning ability of LLMs is not fully explored (as shown in Section 3.1 in the main text). We propose a principal strategy, ‘structure-oriented analysis,’ to activate the self-thinking ability of LLMs and significantly improve the zero-shot reasoning ability, which can reduce the gap between zero-shot reasoning and few-shot reasoning to a large extent. This is different from [1] and [2] mentioned by the reviewer, in which they focus on the design of the agent system rather than improving the capability of the individual LLM.
>
> We further provide theoretical analysis in Section 3.2 to support our strategy. Our findings are different from previous works, as mentioned by the reviewer, which mainly rely on post-reasoning feedback, including internal feedback, such as self-correction, and external feedback, such as multi-agent debate.
>
> Given the observations in Section 3, the proposed agent system in Section 4 is then an additional contribution that provides a more comprehensive and practical solution for the structure-oriented analysis. As mentioned in Section 4.1, Refine Agent is designed to ensure the guidance of structure analysis is well followed; Retrieve Agent is designed to obtain external knowledge to better solve sub-questions from Reason Agent; Memory Agent is designed to track structure analysis and reasoning trajectory. Therefore, the whole design of our agent system is based on our principal strategy, which is different from that of existing agents. The components or architectures of the agent may be similar to existing works, but the main purpose and logic of design is to serve our principal strategy better. We believe our findings and proposed strategy are novel and necessary to understand and unleash the LLM’s reasoning capability.
>
> We have revised the introduction to clarify our main contribution.
>
> Besides, we include a discussion on the agent works mentioned by the reviewer and compare them with ours. [1] introduces an LLM multi-agent framework including an LLM Decomposer, LLM planner, and LLM interface to conduct tasks and interact with the environment in Minecraft. [2] categorizes most of existing LLM agents via information storage, action space and decision-making procedure. [3] focuses on tool-use of LLMs and trains a series of models with enhanced ability of tool-use. [4] proposes an agent system to implement Monte Carlo Tree Search with the help of few-shot examples. We acknowledge that existing works have adopted components like reasoning core, refinement, memory, external knowledge, etc in the agent system.
>
> We do not claim that we develop novel components or structures in agent design. In fact, we adopt the basic components and structures to build a multi-agent system, and our goal is to utilize agents to fully illustrate the effectiveness of the proposed strategy, and structure analysis, rather than proposing a novel agent system.
>
> [1] Ghost in the Minecraft: Generally Capable Agents for Open-World Environments via Large Language Models with Text-based Knowledge and Memory
>
> [2] Cognitive Architectures for Language Agents
>
> [3] ToRA: A Tool-Integrated Reasoning Agent for Mathematical Problem Solving
>
> [4] Language Agent Tree Search Unifies Reasoning, Acting, and Planning in Language Models

---

> > ### Comment · Reviewer_N64Z · 2024-11-26
> >
> > I thank the author for their rebuttals, especially with these large numbers of experiments and revisions.
> >
> > From the rebuttals and revisions, I can see the proof of the method's generality on knowledge-intensive/math/commonsense reasoning tasks and the overall cost efficiency.
> > I think the authors have done a great job on solving my concerns and the revised version does improve the clarity of the work and the purpose.
> > I will raise my score.
> >
> > To make things even better, I think the author should include a formal definition (can be in natural language) in Section 3 to first elaborate on what is the structure-oriented analysis, especially how they differ from normal problem decomposition, for example, maybe a clearer decomposition guideline or leveraging the syntax and commonsense ability of the LLM.
> >
> > Some questions, just to discuss
> >
> > 1. Will SARA work well on small models, such as a 7b model?
> > Current structure-oriented analysis seems to largely depend on the extraction performance of the LLM.
> >
> > 2. Why SARA fails to perform well on MATH?

---

> > > ### Author Response · Authors · 2024-11-27
> > >
> > > We thank the reviewer for the appreciation of our work and further comments!
> > >
> > > 1. In the following, we provide more information on the structure-oriented analysis and comparison with normal decomposition.
> > >
> > > To formally define our proposed structure-oriented analysis, it consists of the following steps (we explain the process with the help of the example in Figure 1 in the paper):
> > >  - Identify key components: Identify the crucial elements (for plain text questions) and variables (for mathematical questions) that play significant roles in the main problem by analyzing the grammar structure of problem statements. They are usually subjects, objects, and attributes in the sentences.
> > > - Relationship between components: Given the key components, explain how they are related to each other following the grammar/syntax structure. For example, if a component is an attribute of another component, they are naturally related.
> > > - Sub-question decomposition: Break down the problem into sub-questions according to the components and their relationships. This can ensure each of the sub-questions focuses on a specific aspect necessary for understanding the solution.
> > > - For each sub-question, describe how solving it helps address the main problem. This helps reduce potential hallucination and aligns sub-questions with analysis in steps 1 and 2.
> > >
> > > Take the example in Figure 1 as an illustration. The original question is ‘What is the name of the fight song of the university whose campus is in Lawrence, Kansas, and whose branch campuses are in the Kansas City metropolitan area?’ This is a complex statement with clauses.
> > > - Step 1 Identify key components: Fight song, University, Main campus location (Lawrence, Kansas), Branch campuses location (Kansas City metropolitan area). These components are subjects, attributes, and subjects in the relative clauses.
> > > - Step 2 Relationship between components: 1. The fight song belongs to the university; 2. The university has a main campus and branch campuses; 3. The main campus is located in Lawrence, Kansas; 4. The branch campuses are located in the Kansas City metropolitan area. These relationships are extracted from the main clause (fight song of the university ) and relative clauses (campus is in xxx, branch campuses are in xxx).
> > > - Step 3 Sub-question decomposition: 1. Which university has its main campus in Lawrence, Kansas? 2. Does this university have branch campuses in the Kansas City metropolitan area? 3. What is the fight song of this identified university? These questions are obtained from analysis in Step 2.
> > > - Step 4 Implication of sub-questions: solving sub-question 1 will provide candidates for the possible universities; solving sub-question 2 will identify the exact university satisfying the constraints of locations; solving sub-question 3 will give the final answer.
> > >
> > > To compare with commonly used prompting strategies, we use least-to-most prompting [1] as a reference method. In [1], their focus is on decomposing the main problem into a sequence of sub-questions with examples provided as demonstration. In our method, no example for decomposition is provided, but instead, we ask the LLM to identify critical components and analyze the relationship between the key components with the help of grammar structures as we described above. Then the sub-questions can be naturally obtained from the analysis. In this case, we get rid of examples and make our method more generalizable.
> > >
> > > Due to the page limit, we have not integrated the above discussion in the revision. We will distill a more concise formal definition and put it in the revised paper; the above illustration on an example question may be organized in the appendix.
> > >
> > > 2. We agree that the analysis relies on the model's capability of semantic understanding and extraction. We are running experiments on smaller models, Llama3-8B, and will update once they are finished.
> > >
> > > 3. For the performance of the MATH dataset, as discussed in Section 5.3 of the revised paper, the MATH dataset contains some problems solely expressed in mathematical symbols that do not exhibit clear structures like natural language. Therefore, our analysis struggles to extract as much information as those problems stated in natural language, and the effect of structure analysis is restricted. We note that it may need additional strategies, such as fine-tuning symbolic datasets, to enhance the model's capability of understanding symbolic structures.
> > >
> > > [1] Zhou, Denny, et al. Least-to-most prompting enables complex reasoning in large language models.

---

### Official Review · Reviewer_8je1 · 2024-11-02

**Soundness:** 4
**Presentation:** 3
**Contribution:** 3
**Rating:** 8
**Confidence:** 4

**Summary:**

The authors introduce a structure-oriented analysis method that enhances LLMs’ ability to understand question structure by systematically identifying and reasoning through key components before addressing a problem. This structured approach, applicable to most standard LLM prompting methods, is validated with CoT and ReAct, using syntactic and grammatical analysis to identify crucial components, relationships, and sub-questions within a problem statement.
To implement this structure-oriented analysis, the authors uses four components—Reason, Refine, Retrieve, and Shared Memory—that collectively enforce structured reasoning, improve accuracy through iterative refinement and retrieval of external knowledge.

The theoretical foundation for this approach is grounded in probabilistic graphical models (PGMs), which the authors use to illustrate how structure-oriented analysis can guide LLMs along correct reasoning paths. By capturing the relationships between observed knowledge components and latent variables, PGMs demonstrate how this structured approach minimizes reasoning errors by identifying critical intermediate variables along the optimal reasoning path.

Experiments show that SARA effectively enhances performance across diverse natural language tasks, with results sometimes surpassing few-shot methods. Additionally, SARA exhibits robustness against attacks aimed at disrupting the reasoning process.

**Strengths:**

- Interesting theory motivation
- Really good experimental protocol:
    - Ablations on the contributions of each part of the structure, retrieval and refinement are complete to
    - Showed the versatility to multiple base models

**Weaknesses:**

- Only evaluated the application of the method to natural language tasks. Would have liked to see how that applies to other tasks that required reasoning, e.g. math. There’s probably not a 1:1 mapping with grammar/syntax, but task decomposition seems quite close, and I see your method general enough to do that. Have you already tried this?
- The correctness/soundness structured analysis is only captured by the downstream performance on the task. Have you thought about investigating the analysis in a more direct way? Would the use of structured outputs (e.g. parsing the string into 1.,2.,3.,4. enable you to create evaluation metrics?)
- Robustness on attacks is interesting theoretically but i) I am not sure how important that is to the story, ii) attacks on few-shot: you also show that the method is applicable to few-shots prompting.
- I don’t see a limitation paragraph, what are those? Can you add a paragraph?

**Questions:**

- The structured oriented analysis seems to be quite close to [1], but not grounded on reasoning modules (which is nice).
- In practice, I am curious how such attacks could happen as they would be happening on backend side, whereas easier prompts attacks could happen on client side.
- “However, these approaches either rely on task-specific examples (few-shot) or suffer from ineffectiveness on complex tasks (zero-shot).” —> you mentioned ToT and GoT. How do they compare in accuracy on the benchmarks you ran (not asking for running this baseline, but reporting if there are numbers in their paper) but to have an idea on their ‘ineffectiveness’

[1] 'Self-Discover: Large Language Models Self-Compose Reasoning Structures', Pei Zhou et al.

---

> ### Author Response · Authors · 2024-11-21
> **Formal response 1/**
>
> We appreciate the reviewer for sharing useful comments to improve our manuscript. We provide the response to your comments as follows:
>
> 1. **[W1] Additional experiments on diverse tasks and datasets.**
>
> The additional experiments as in the following table (copied from the global response):
>
> | | Dataset|Vanilla| ICL(6-shot)|CoT(6-shot)|ReAct(6-shot)|CoK(6-shot)|CoT-SC@10(0-shot)|SARA|
> |-|-|-|-|-|-|-|-|-|
> |GPT4|GSM8K|66.8%|66.9%|92.1%|93.7%|91.9%|87.8%|94.2%|
> | |MATH|43.1%|55.4%|69.2%|67.5%|68.6%|64.1%|68.2%|
> | |StrategyQA|65.6%|68.1%|82.9%| 81.7%| 83.2%| 81.4%| 86.4%|
> |Qwen-max|GSM8K|68.6%|72.8%|87.5%|89.2%|87.6%|84.2%|91.3%|
> | |MATH|42.8%|45.6%|64.9%|64.5%|65.3%|61.9%|64.7%|
> | |StrategyQA|73.4%|75.5%|89.6%|88.4%|90.5%|83.1%|90.7%|
> |Qwen2-57B|GSM8K|54.9%|59.2%|82.7%|83.9%|83.5%|74.5%|84.4%|
> | |MATH|30.1%|33.5%|46.2%|47.3%|46.8%|40.8%|46.5%|
> | |StrategyQA|58.4%|63.2%|85.1%|89.2%|88.3%|79.1%|91.5%|
> |Llama3-70B|GSM8K|55.3%|58.3%|83.7%|86.5%|87.2%|76.8%|89.7%|
> | |MATH|30.7%|32.4%|42.9%|46.3%|44.9%|36.4%|44.2%|
> | |StrategyQA|57.9%|65.1%|84.2%|85.2%|85.8%|80.5%|87.1%|
>
> We notice that our method works well on GSM8K. Since these questions in GSM8K are described in natural language, our strategy can still provide a deeper understanding of the question and lead to the correct solution. On the other hand, MATH dataset contains symbolic reasoning problems and we do notice that SARA does not outperform baselines on this dataset. This may suggest that our strategy does not work well on questions fully expressed with symbols. We highlight this limitation in the revision Section 7, and consider it our future direction to improve.
>
> 2. **[W2] Elaborate on the evaluation of structure analysis**
>
> We thank the reviewer for raising this valuable comment. Since the major goal of this work is to improve the zero-shot reasoning ability of LLMs, we use prediction correctness as the evaluation metric. To confirm that structure analysis indeed helps, we conduct ablations, as shown in Figure 2. While we agree that evaluating how correct the structure-oriented analysis is can provide more details about the proposed method, this evaluation does not directly measure whether the proposed method can benefit reasoning or not. Besides, it is also expensive to develop a dataset for this purpose as it may require extensive human annotation efforts to manually analyze the structure. As a result, we directly test the reasoning performance to align with the main purpose of our work.
>
> 3. **[Q1] Discussion on the mentioned work.**
>
> We thank the reviewer for mentioning this work [1]. This work proposes prompting the LLM to select proper reasoning modules first and then adapting them to each question for better solving. This method combines various reasoning strategies and few-shot examples to improve performance. Our work focuses on the zero-shot reasoning capability of LLMs and the proposed structure analysis is not grounded on reasoning modules. We include this work in the revision (Section 2).
>
> 4. **[W3, Q2] Elaborate on robustness**
>
> We include the section on robustness to show that our method is robust on potential attacks and to support our claim that the structure analysis can extract key concepts and relevant information and filter out irrelevant information. Existing attacks on reasoning usually insert irrelevant and malicious information in the problem or reasoning steps (in few-shot examples) to corrupt the performance. The robustness experiments show that our method can identify the semantic and grammar structures and ignore irrelevant information. These results demonstrate the advantage of the proposed method.
>
> In our experiments, one of the attacks targets zero-shot reasoning (Preemptive attack) while another (Badchain) target few-shot reasoning. We include them for a more comprehensive evaluation, and this does not mean our method is a few-shot method.To adopt Badchain on our method, we simply replace the original problem with the problem attached with the trigger. We have revised our manuscript and updated the clarification in Section 5.6.
>
> In terms of how these attacks work, they assume that the potential attacker provides third-party prompt engineering services and has access to the victim’s prompts. We directly utilize their implementation in our experiments.
>
> 5. **[W4] Limitation paragraph**
>
> Based on your comment, we have updated a new Section 7 to discuss the limitations of the proposed method. Briefly speaking, there are two major limitations. (1) Due to the nature of the structure-oriented analysis, our current method works better on problems that are clearly described in natural languages (e.g., GSM8K) than tasks with pure symbol expressions (e.g., MATH). It can be overcome by extracting the symbolic expressions. (2) Our current agent framework only uses the Retrieve Agent instead of other possible agents with additional tools. Adding more agents can enable the framework doing more tasks, e.g., executing code or a calculator.

---

> > ### Author Response · Authors · 2024-11-21
> > **Formal response 2/**
> >
> > 6. **[Q3] Discussion on ToT and GoT**
> >
> > The original paper of ToT and GoT does not include experiments on datasets used in this work, and we will try to provide ToT and GoT results if time allows. ToT has experiments on GSM8K and StrategyQA as follows (on GPT-4):
> >
> > |Dataset|Acc|
> > |------------|-----|
> > | GSM8K      | 90% |
> > | StrategyQA | 83% |
> >
> > According to the results, ToT has similar performance with CoT on these datasets.
> > We would like to also clarify more about ToT and GoT. ToT and GoT may not need task-specific examples but they rely on task-specific prompting as shown in [1], [2](Appendix C D E). Therefore, by the sentence mentioned by the reviewer, we do not mean they are ineffective but mean that they are task-specific as those few-shot methods. We update in revisions (Section 2) to avoid such confusion.
> >
> > [1] https://github.com/princeton-nlp/tree-of-thought-llm/tree/master/src/tot/prompts
> >
> > [2] Graph of Thoughts: Solving Elaborate Problems with Large Language Models

---

> > ### Comment · Reviewer_8je1 · 2024-11-21
> > **Answer to authors**
> >
> > Thanks for your answer and complete work. I will be keeping my score.
> > Is it also possible to add to the appendix the structured analysis on Hendrycks MATH? Also GPT4 at 43.1 seems a bit off?

---

> > > ### Author Response · Authors · 2024-11-22
> > >
> > > We thank the reviewer for further comment. We include examples for both GSM8K and MATH in the Appendix H for reference.
> > >
> > > As for the performance of GPT-4 on MATH dataset, we would like to clarify that 43.1% is the performance when we directly ask the model to solve the question without any prompting techniques like CoT, and this result is consistent with the reported number 42.2% in Table 1 of [1]. We would like to further provide 0-shot CoT as a supplementary. These results are also consistent with existing works and those mentioned by the reviewer.
> > >
> > > | Dataset | Vanilla | CoT(0-shot) |
> > > |---------|---------|-------------|
> > > | GSM8K   | 66.8%   | 84.3%       |
> > > | MATH    | 43.1%   | 63.6%       |
> > >
> > > [1] Solving Challenging Math Word Problems Using GPT-4 Code Interpreter with Code-based Self-Verification (https://openreview.net/forum?id=c8McWs4Av0)

---

### Official Review · Reviewer_i9ee · 2024-11-02

**Soundness:** 3
**Presentation:** 3
**Contribution:** 2
**Rating:** 3
**Confidence:** 4

**Summary:**

The paper proposes using structure-oriented analysis to improve zero-shot reasoning of language models (specifically for knowledge-intensive tasks). The authors motivate how structure-oriented analysis by leaning on the hypothesis that inference with language models emulates a PGM constructed on the pretraining data. The models explore neighboring nodes in the PGM based on key properties of the question, and the knowledge of states on the path of correct reasoning helps reduce reasoning errors. Then the authors get into their method SARA, which comprises three agents performing reasoning, refinement, and retrieval. Their results show that their method improves performance over baselines such as CoT and ReAct (zero-shot) on tasks like hotpotQA, fever, MMLU.

**Strengths:**

* The paper is well-organized and easy to follow
* Their system SARA achieves reasonable improvements over other prompting baselines
* The PGM analysis is intuitive and could potentially be useful to the larger community

**Weaknesses:**

* The paper is missing a lot of related works that focus on sub-task or sub-question decomposition to improve reasoning or LLM agent performance, see the list below:
    * [Socratic CoT](https://arxiv.org/abs/2212.00193)
    * [Least to Most Prompting](https://arxiv.org/abs/2205.10625)
    * [Decomposed Prompting](https://arxiv.org/abs/2210.02406)
    * [Adapt](https://arxiv.org/abs/2311.05772)
    * [Screws](https://arxiv.org/abs/2309.13075)
    * [ART](https://arxiv.org/abs/2311.07961)
    * [Self-Discover](https://arxiv.org/abs/2402.03620)

*  The primary assumption in this work (corroborated by ablations) is that *syntactic structures are useful to guide the reasoning process*, which seems very specific to knowledge-intensive multi-hop QA and not just any reasoning dataset. For e.g., I don't see a reason why this setup should work for math or logical reasoning datasets (GSM8K, MATH, ARC, BigBench, etc.) To that end, I found the title and abstract overly generic or a bit misleading since the scope of the work does not cover all forms of reasoning.

* Are the main improvements coming from the fact that instead of one environment (retrieval) in ReAct, which is already quite expensive, we are increasing the computational budget of each step (reason, retrieve, verify/refine)? I would like to see how SARA fairs against self-consistency, ReAct with multiple trials, or step-wise self-consistency for agentic tasks as in [this paper](https://arxiv.org/abs/2402.13212)

* On a related note,  the paper lacks a cost (token budget) vs. performance tradeoff. SARA generates far more tokens than any of the baselines; comparison with just a few-shot CoT/ReAct may not suffice since additional tokens are on the input side. Why not have zero-shot SC as a baseline, or sample multiple responses and ask the model to decide which one is best / [Meta-reasoning](https://arxiv.org/abs/2304.13007)

**Questions:**

1. Are there any more datasets where this method would find application?
2. The paper hinges on the model discovering and generating "structures." Is it fair to rely on the model's ability to do so without any training?  How will this generalize to other tasks where finding such structures is harder, like for math reasoning?
3. What is the role of the memory in the agent? Is the memory instance specific, if so how is it different from just keeping history around in ReAct trajectory?
4.  Why not use EM (exact match) for HotpotQA like the ReAct paper, what is the justification for this choice, and what do the results look like with EM metric for HotpotQA?

---

> ### Author Response · Authors · 2024-11-21
> **Formal response 1/**
>
> We thank the reviewer for reviewing our manuscript. Below are the responses and clarifications towards your comments:
>
> 1. **[W1] Discussion on related works**
>
> Thanks for sharing the list of related works. We have checked these works. Below is a summary of the comparisons with them.
>
> [1] proposes SOCRATIC COT, which trains a generator to decompose the question and a question-answering model to solve these sub-questions. Their method is designed to utilize CoT examples generated by the large models to improve the reasoning ability of small models. Therefore, their method has a different scope as we focus on a general strategy for zero-shot reasoning capability. Besides, SOCRATIC COT requires additional training.
>
> [2] proposes the least-to-most prompting technique to query the model to decompose the problem first and then solve sub-questions. However, they rely on few-shot examples and could be considered as a few-shot reasoning method, while we focus on the zero-shot method and leverage the model’s ability to analyze semantic and grammar structures.
>
> [3] has a similar strategy to [1] and proposes a decomposer to decompose the problem into sub-tasks. However, it also relies on few-shot examples in decomposing and sub-task solving, which differs from this work’s focus.
>
> Similarly, [4] also decomposes the task first using a planner, which is powered by an LLM, and then utilizes executors to solve these tasks. This framework, ADAPT, also requires task-specific examples to help decompose and execute, as shown in its appendix.
>
> [5] proposes a modular framework, SCREWS, to enhance reasoning with revisions. SCREWS will first generate an initial output and then a refined output. Two outputs will be sent to the selection module to choose the correct one. Their module design can accommodate existing methods like CoT and self-consistency. However, our primary goal is not to propose a reasoning system but to explore the limit of the zero-shot reasoning capability of LLMs. So, we observe and propose the structure analysis to activate the self-thinking ability of LLMs and enable them to automatically reason on questions without task-specific prompting or examples.
>
> [6] proposes a refinement strategy (ask, refine, trust) to reduce the potential error during the reasoning process. While we adopt a similar strategy by leveraging a Refine Agent to evaluate and refine the reasoning step, we aim to ensure the reasoning trajectory follows the structure analysis as we mentioned in Section 4.1. We do not claim the design of Refine Agent to be our novel contribution, but it is a necessary strategy to unleash the power of structure analysis fully, which is the core of this work.
>
> [7] proposes prompting the LLM to select proper reasoning modules first and then adapting them to each question for better solving. This method combines various existing reasoning strategies and few-shot examples to improve performance. However, we primarily focus on exploring the limit of zero-shot reasoning capability rather than proposing a reasoning system.
>
> Due to page limit, we summarized the above and updated a shorter version in Section 2.
>
> [1] Distilling Reasoning Capabilities into Smaller Language Models.
>
> [2] Least-to-Most Prompting Enables Complex Reasoning in Large Language Models
>
> [3] Decomposed Prompting: A Modular Approach for Solving Complex Tasks
>
> [4] ADaPT: As-Needed Decomposition and Planning with Language Models
>
> [5] SCREWS: A Modular Framework for Reasoning with Revisions
>
> [6] The ART of LLM Refinement: Ask, Refine, and Trust
>
> [7] Self-Discover: Large Language Models Self-Compose Reasoning Structures
>
> 2. **[W2, Q1] Diverse tasks and datasets**
>
> Thanks for your comment, we have updated some additional experiments for different tasks and datasets as in the following table (copied from the global response):
>
> | | Dataset|Vanilla| ICL(6-shot)|CoT(6-shot)|ReAct(6-shot)|CoK(6-shot)|CoT-SC@10(0-shot)|SARA|
> |-|-|-|-|-|-|-|-|-|
> |GPT4|GSM8K|66.8%|66.9%|92.1%|93.7%|91.9%|87.8%|94.2%|
> | |MATH|43.1%|55.4%|69.2%|67.5%|68.6%|64.1%|68.2%|
> | |StrategyQA|65.6%|68.1%|82.9%| 81.7%| 83.2%| 81.4%| 86.4%|
> |Qwen-max|GSM8K|68.6%|72.8%|87.5%|89.2%|87.6%|84.2%|91.3%|
> | |MATH|42.8%|45.6%|64.9%|64.5%|65.3%|61.9%|64.7%|
> | |StrategyQA|73.4%|75.5%|89.6%|88.4%|90.5%|83.1%|90.7%|
> |Qwen2-57B|GSM8K|54.9%|59.2%|82.7%|83.9%|83.5%|74.5%|84.4%|
> | |MATH|30.1%|33.5%|46.2%|47.3%|46.8%|40.8%|46.5%|
> | |StrategyQA|58.4%|63.2%|85.1%|89.2%|88.3%|79.1%|91.5%|
> |Llama3-70B|GSM8K|55.3%|58.3%|83.7%|86.5%|87.2%|76.8%|89.7%|
> | |MATH|30.7%|32.4%|42.9%|46.3%|44.9%|36.4%|44.2%|
> | |StrategyQA|57.9%|65.1%|84.2%|85.2%|85.8%|80.5%|87.1%|
>
> (To be continued in the following response)

---

> ### Author Response · Authors · 2024-11-21
> **Formal response 2/**
>
> (Continued from the previous response)
>
> In short, our method performs the best in GSM8K and StrategyQA compared to other methods and achieves a comparably good performance in MATH. We conjecture that the structure-oriented analysis by its nature focuses more on the language, thus it can better understand the question in GSM8K than the other methods, but its performance in MATH is not as superior as in the other tasks.
>
> For MATH dataset, to explain why SARA still improves performance, first, there are also some problems described in natural language, and SARA can understand the language better. Second, the combination of step-wise reasoning and refinement in SARA also guarantees its effectiveness. Therefore, SARA can still work better than 0-shot methods, and it is an interesting future direction to study how to better handle symbolic representations in math problems.
>
> 3. **[W3, 4] discussion of computation cost.**
>
> The statistics of the computational cost are summarized in the following table (copied from the global response):
>
> | |HotpotQA| |FEVER| |
> |-|-|-|-|-|
> | |Input|Output|Input|Output|
> |ReAct|1632|451|862|338|
> |CoK|791|379|587|291|
> |CoT-SC@10(0-shot)|276|2249|85|2057|
> |SARA|462|716|476|599|
>
>  In short, our method is affordable compared with baselines, and achieves a good balance between the cost of tokens and effectiveness. We also include zero-shot CoT with self-consistency as an additional zero-shot baseline on all tasks and datasets. Results are updated in Table 1, 2 and Table 8 in Appendix G.
>
> 4. **[Q2] Clarification on the structure analysis**
>
> As mentioned in Section 1, existing works have shown that LLMs show a strong ability to understand and parse semantic and grammar structures in the text. Therefore, we directly leverage this ability in our work. As demonstrated in the new results, this ability generalizes well to math problems that are written in natural language, such as GSM8K dataset, where SARA shows superior performance. However, for some problems written with math symbols, such as some equation-solving problems in MATH dataset, there are few grammar or semantic structures to capture, thus the performance of SARA is not as superior as in other tasks. These results together with our original results indicate that our method generalizes well on problems that are described in natural language, and we agree that it may need some modifications such as additional training to generalize to symbolic problems. We mention this limitation in the revised Section 7 and take it as an interesting future direction.
>
> 5. **[Q3] Elaborate on memory**
>
> Our memory is designed to be instance-specific and store everything in the reasoning process, including structure analysis, reasoning trajectory, and retrieved knowledge. The difference between our method and ReAct is that we have multiple agents, so we need this memory module as a shared message pool with which all agents can interact. Besides, different from ReAct where all previous thoughts and observations are directly fed into the model, we only send part of the memory into the agent. For example, the Refine Agent in SARA only needs one previous step and corresponding knowledge. These highlight the necessity of the memory module and the difference between our design and ReAct.
>
> 6. **[Q4] Elaborate on metric EM**
>
> We notice that, unlike Fever and MMLU, whose answer is covered by some options (like True/False, A/B/C/D), the model-generated answer for HotpotQA is usually in free-form, such as ‘George Archainbaud died first than  Ralph Murphy’ and ‘Pirate's Cove was published more recently’ and hard to extract the answer from them using some general hard rules. Thus, using EM can cause false positives and false negatives, especially for zero-shot methods (Vanilla, SARA), where no examples are provided to regularize the form of the final answer. Therefore, we leverage LLMs to compare the generated and ground-truth answers.

---

> > ### Comment · Reviewer_i9ee · 2024-11-21
> >
> > I thank the authors for their efforts. However, I am still not satisfied with the modifications due to the following reasons:
> >
> > - There is no experimental setup for the math reasoning datasets in their paper, and the author's reported numbers for GPT-4 on MATH and GSM8K appear to be alarmingly under the reported values (see: https://huggingface.co/spaces/allenai/ZeroEval, https://ai.meta.com/blog/meta-llama-3-1/, etc). This may be due to a bug in extracting and matching answers for MATH and GSM8K, making the new results unreliable.
> > - Is the vanilla baseline directly answering the question without a chain of thought? If so there should be a 0-shot CoT baseline as well.
> > - As far as least to most prompting is considered, I find it hard to imagine that if the models can follow detailed instructions as in the prompt in appendix B for SARA, which includes sub-question decomposition, why the authors cannot design an instruction that asks the model to decompose subquestions and answer them one after another as in the LtM paper.
> > - Can the authors include any qualitative examples of the outputs generated by different components of SARA for MATH and GSM8K, so that we can verify that it is indeed the "structures" in language that are responsible for performance gains?

---

> > > ### Author Response · Authors · 2024-11-22
> > >
> > > We thank the reviewer for further comments.
> > >
> > > - We would first clarify that Vanilla in the baseline is the performance when we directly ask the model to give an answer to the question **without any reasoning-enhancing prompting techniques**.
> > > - We would like to clarify that our results are consistent with GPT-4 Technical Report [1] and other existing works such as [2]. In particular, for GSM-8K, in Table 2 on Page 7 of [1], the accuracy is **92.0%** using **5-shot CoT**. To compare with, the 6-shot CoT in our experiment is 92.10%. In terms of MATH, in Table 1 on Page 8 of [2], it clearly states that **without CoT and other techniques**, GPT-4 only achieves **42.2%** accuracy, and in our experiment, this number is **43.10%** for the vanilla method (directly ask the model to answer the question without CoT or other techniques). Our results align with these existing works.
> > > - We also would like to mention that the detailed description of the baselines used in experiments can be found in Section 5.1 and the datasets in Appendix D. Based on your suggestion, we further provide 0-shot CoT as a supplementary. These results are also consistent with existing works and those mentioned by the reviewer.
> > >
> > > | Dataset | Vanilla | CoT(0-shot) |
> > > |---------|---------|-------------|
> > > | GSM8K   | 66.8%   | 84.3%       |
> > > | MATH    | 43.1%   | 63.6%       |
> > >
> > > - As for the least-to-most prompting, we do notice that its performance is not optimal as mentioned in [2] (sometimes worse than CoT). This can be because sub-questions are not extracted accurately or not sufficient to solve the problem. We would also like to clarify that decomposing sub-questions is not the primary goal of this work but a by-product of structure analysis. Therefore, we do not strictly ask the model to answer these sub-questions, but conduct conditional reasoning based on analysis which can combine the advantages of both decomposition and flexible reasoning.
> > > - We have updated our revision and included examples for both GSM8K and MATH in the Appendix H for reference.
> > >
> > > [1] GPT-4 Technical Report (https://arxiv.org/pdf/2303.08774)
> > >
> > > [2] Solving Challenging Math Word Problems Using GPT-4 Code Interpreter with Code-based Self-Verification
> > > (https://openreview.net/forum?id=c8McWs4Av0)

---

> > > > ### Comment · Reviewer_i9ee · 2024-11-22
> > > >
> > > > In that case, you must include 0-shot CoT a baseline to your zero-shot method in the main paper (for it to be sound). Additionally, I am unsure of your implementation of CoT SC@10 (0-shot) number as compared to the greedy 0-shot baseline you are only getting 1-3% improvement, when most papers and in my experience too is much more, including [2] that you cite. Also please state which GPT-4 model you are using.
> > > >
> > > > I am skeptical of the veracity of your additional experiments and baselines now.

---

> > > > > ### Author Response · Authors · 2024-11-24
> > > > >
> > > > > We thank the reviewer for the further comment.
> > > > >
> > > > > We agree with the reviewer’s suggestion to add 0-shot CoT as a baseline. We are running experiments on all datasets and models for a comprehensive illustration. We will update in revision and notify all reviewers as soon as we finish.
> > > > >
> > > > > We tried our best to search the literature but could not find the results about 0-shot CoT with self-consistency (not a few-shot CoT-SC) using GPT4 on GSM8K and MATH datasets. We do notice that there exist factors influencing the performance such as the number of CoT solutions for voting, temperature parameters, and implementation of majority voting. We would be grateful if the reviewer could provide some suggested literature so that we can compare. We are willing to adopt the same setups and make a fair comparison. In addition to the 0-shot CoT with self-consistency, it would also be great if the reviewer could provide other related literature of interest so that we can thoroughly compare our results with them to validate the effectiveness of our approach. We would like to clarify that we only implement Vanilla, ICL and CoT on our own as they are simple baselines. All other baselines (ReAct, CoK) are implemented with the code provided by the original papers.
> > > > >
> > > > > Our code is also open-sourced and we attach an anonymous GitHub repo in the revision Section 5 (https://anonymous.4open.science/r/ReasonAgent-4458). More specifically, our implementation of CoT_SC can be found in https://anonymous.4open.science/r/ReasonAgent-4458/retrieve_reason/CoT_SC.py , running by ``python CoT_SC.py --model MODEL --num NUM_OF_CoT --dataset DATASET``
> > > > >
> > > > > Moreover, we do have (indirect) evidence to show that our implementation works properly. We notice a paper [1] also leveraging an LLM to implement majority voting in self-consistency. So we compare our results with their reported ones fairly. Specifically, we follow the same parameter setup, including testing on GPT-3.5-turbo, generating  8 CoT solutions, leveraging the LLM for majority vote, and the temperature is set as 1.0. We test on both GSM8K and MATH and present both our results and the number reported in [1] as follows. There are some (potential) implementation differences from [1]: First, they adopt a few LLM-generated examples as few-shot examples while we do not have examples. Second, since [1] does not mention the exact version of GPT-3.5-turbo, the model may be of a different version than ours.
> > > > >
> > > > > |       | Our  implementation | [1] USC(Table 4) |
> > > > > |:-----:|:-------------------:|:----------------:|
> > > > > | GSM8K |        81.9%        |       77.8%      |
> > > > > |  MATH |        40.4%        |       38.1%      |
> > > > >
> > > > > While our results are slightly better than those in [1], the performance is similar, which indicates that our implementation and evaluation are proper and reliable.
> > > > >
> > > > > [1] Universal Self-Consistency for Large Language Models

---

> > > > > > ### Author Response · Authors · 2024-11-27
> > > > > >
> > > > > > We are grateful for the reviewer's valuable comments. We have updated the revision to include 0-shot CoT as a baseline.

---

> > > > > > > ### Author Response · Authors · 2024-11-29
> > > > > > > **A friendly reminder**
> > > > > > >
> > > > > > > We are grateful for the reviewer's valuable comments. Since the rest time of the discussion period is limited now, please let us know if you have other concerns. We are looking forward to hearing back from you.

---

> > > > > > > > ### Author Response · Authors · 2024-12-01
> > > > > > > > **A friendly reminder**
> > > > > > > >
> > > > > > > > We sincerely thank you for your valuable comments. We have diligently worked to address the concerns you raised and hope our responses are satisfactory. As the discussion period is nearing its end, we kindly request any additional feedback you might have. If you have any further questions or require additional clarification, please do not hesitate to let us know.

---

### Official Review · Reviewer_57Cm · 2024-11-03

**Soundness:** 3
**Presentation:** 3
**Contribution:** 2
**Rating:** 3
**Confidence:** 4

**Summary:**

This paper stressed the challenges of zero-shot reasoning ability of LLMs on complex tasks. Inspired by human reasoning behaviros, the authors introduce a method based on the linguistic and logical structures to help LLMs break down complex tasks and give the answer through a iterative reasoning process. The authors develop a multi-agent system SARA based on the method with a reason agent to analyze grammar and syntax, a refinement agent to resolve inconsistencies and logical error, a retrival agent to access external knowledge and a shared memory to store the intermediate states. Compared with few-shot baselines, the authors performed experiments on four models across four different datasets to validates the effectiveness of the method. In addition, the method shows robustness against malicous injections in demonstrations and irrelevant information in problem statements.

**Strengths:**

1. This work introduce a intuitive method to break down complex tasks through grammar and syntax structure analysis and help LLMs to give answers via iterative reasoning.
2. This work provides theoretical analysis and empirical evidence to validate the effectiveness of the method. The experiments cover both open-source and closed-source models and are performed on both literal and science tasks. All the models show remarkably higher performance than  baselines.

**Weaknesses:**

1. The authors have conducted ablation studies on each components of the SARA system. However, the experiments are limited to agents that backended by the same model. To enhance the study, it is recommended to perform future experiments including analyses examining the impact of assigning different models to different agent roles within the system and how the models' performance influences each parts.
2. The method is somewhat incremental considering many previous works about multi-agent systems/debate systems and the novelty is limited.

**Questions:**

1. The performance of Llama3-70B on MMLU-BIO and MMLU-PHY improves significantly using SARA compared to its vanilla results but shows little advantage compared to GPT-4. Why can't this method help open-source LLMs out-performs closed-sourced ones on STEM tasks?
2. I wonder the effectiveness of this method on math reasoning problems.

---

> ### Author Response · Authors · 2024-11-21
> **Formal response 1/**
>
> We appreciate the reviewer in providing constructive suggestions and useful comments. In addition to the new results and updates in the revision, below are the itemized responses to your comments:
>
> 1. **[W1] Additional ablation experiments**
>
> We thank the reviewer for this suggestion. In our paper, we presented the ablation study regarding components in the structure-oriented analysis and the combinations of agents in SARA. Those experiments can be considered as extreme cases where the LLMs are entirely removed from the corresponding parts (structure-oriented analysis components and agents in SARA). We conjecture that replacing the LLM in those parts should lead to performance between the full settings (best case) and the one of which the corresponding part is removed in the ablation study (worst case).
>
> To verify our conjecture, we are conducting experiments that a stronger model (GPT-4) is replaced by a smaller model (Qwen2-57B) for agents to illustrate the influence of models. We will update the results once finished.
>
> 2. **[W2] Elaborate on novelty**
>
> The key contributions of our work are around the theme of structure-oriented autonomous (zero-shot) reasoning, which provides insights into general mechanism design and theoretical support for LLM-based reasoning, and guides our multi-agent system design.  We would like to clarify our major contributions and emphasize why we are different from existing works.
>
> *  We observe that the zero-shot reasoning ability of LLMs is not fully explored (as shown in Section 3.1 in the main text). Therefore, the first contribution is a principal strategy, ‘structure-oriented analysis,’ to activate the self-thinking ability of LLMs and significantly improve the zero-shot reasoning ability, which can reduce the gap between zero-shot reasoning and few-shot reasoning to a large extent.
> * In Section 3, a second contribution in our work is that we further provide theoretical analysis in Section 3.2 to support our strategy. Our findings are different from previous works, as mentioned by the reviewer, which mainly rely on post-reasoning feedback, including internal feedback, such as self-correction, and external feedback, such as multi-agent debate.
> * Given the observations in Section 3, the proposed agent system in Section 4 is then a third contribution that provides a more comprehensive and practical solution for the structure-oriented analysis. As mentioned in Section 4.1, Refine Agent is designed to ensure the guidance of structure analysis is well followed; Retrieve Agent is designed to obtain external knowledge to better solve sub-questions from Reason Agent; Memory Agent is designed to track structure analysis and reasoning trajectory. Therefore, the whole design of our agent system is based on our principal strategy, which is different from that of existing agents. The components or architectures of the agent may be similar to existing works, but the main purpose and logic of design is to serve our principal strategy better. We believe our findings and proposed strategy are novel and necessary to understand and unleash the LLM’s reasoning capability.
>
> To clarify the novelty and major contribution of this paper, we revised the last paragraph of Section 1 on Page 2 to highlight our scientific contribution of the structure-oriented analysis.

---

> > ### Author Response · Authors · 2024-11-21
> > **Formal response 2/**
> >
> > 3. **[Q1] Discussion on the performance gap between open-source and API-only models**
> >
> > We thank the reviewer for this comment.
> >
> > First, we have to acknowledge that the selected open-source models are inferior to API-only models in terms of their original reasoning capability and internal knowledge (as shown in the “Vanilla” column of Table 1) in the tasks of this paper. Such original capability differences can affect the performance of methods with enhancement (i.e., SARA, ICL, CoT, ReAct, etc). This is the fundamental reason for the performance gap between the open-source and API-only models.
> >
> > According to Table 1 in our paper, the performance of SARA on Llama3-70B is comparable to or even better than the performance of Vanilla on GPT-4, which indicates that SARA can fill the gap between open-source models and stronger API-only models. We also notice that “SARA on Llama3-70B” performs worse than “few-shot CoT, ReAct and SARA on GPT-4”. This is because these reasoning methods also improve the reasoning capability of GPT-4. Therefore, we would like to mention that our method can make open-source models perform similarly or even better than API-only models(Vanilla). When applying reasoning-enhancing methods to both kinds of models, the inherent gap between them makes it hard for open-source models to perform better than API-only models.
> >
> > Besides, we would like to clarify the main purpose of this work. We aim to propose a principal strategy to activate the self-thinking capability of LLMs and improve the zero-shot reasoning capability. It is obvious that we indeed reduce the gap between zero-shot reasoning and few-shot reasoning on the same LLMs. Our primary goal is not to achieve better performance with open-source models than with more advanced API-only models.
> >
> > 4. **[Q2] Effectiveness in math problems**
> >
> > Thanks for your comment, we have updated some additional experiments for different tasks and datasets. The results are as follows (the table is copied from the global response):
> >
> > | | Dataset|Vanilla| ICL(6-shot)|CoT(6-shot)|ReAct(6-shot)|CoK(6-shot)|CoT-SC@10(0-shot)|SARA|
> > |-|-|-|-|-|-|-|-|-|
> > |GPT4|GSM8K|66.8%|66.9%|92.1%|93.7%|91.9%|87.8%|94.2%|
> > | |MATH|43.1%|55.4%|69.2%|67.5%|68.6%|64.1%|68.2%|
> > | |StrategyQA|65.6%|68.1%|82.9%| 81.7%| 83.2%| 81.4%| 86.4%|
> > |Qwen-max|GSM8K|68.6%|72.8%|87.5%|89.2%|87.6%|84.2%|91.3%|
> > | |MATH|42.8%|45.6%|64.9%|64.5%|65.3%|61.9%|64.7%|
> > | |StrategyQA|73.4%|75.5%|89.6%|88.4%|90.5%|83.1%|90.7%|
> > |Qwen2-57B|GSM8K|54.9%|59.2%|82.7%|83.9%|83.5%|74.5%|84.4%|
> > | |MATH|30.1%|33.5%|46.2%|47.3%|46.8%|40.8%|46.5%|
> > | |StrategyQA|58.4%|63.2%|85.1%|89.2%|88.3%|79.1%|91.5%|
> > |Llama3-70B|GSM8K|55.3%|58.3%|83.7%|86.5%|87.2%|76.8%|89.7%|
> > | |MATH|30.7%|32.4%|42.9%|46.3%|44.9%|36.4%|44.2%|
> > | |StrategyQA|57.9%|65.1%|84.2%|85.2%|85.8%|80.5%|87.1%|
> >
> >  In short, our method performs the best in GSM8K compared to other methods, and achieves a strong performance in MATH comparable to the best. We conjecture that the structure-oriented analysis by its nature focuses more on the language, thus its performance in MATH is not as superior as in the other tasks. It is an interesting future direction to study how to better handle symbolic representations in math problems.

---

> > > ### Author Response · Authors · 2024-11-27
> > >
> > > We are grateful for the reviewer's valuable comments. We have updated the revision to include additional ablation to handle your concern in weakness 1. Specifically, we replace the GPT-4 in Retrieval Agent and Refine Agent with a smaller model, Qwen2-57B. Results are presented in Figure 5. A smaller model can still handle the task of Retrieval Agent but will lead to performance dropping of the Refine Agent. This suggests that the Refine Agent requires better reasoning capability of models while the Retrieval Agent can utilize smaller models.
> > >
> > > We hope our responses adequately address your concerns. We are looking forward to your feedback.

---

> ### Author Response · Authors · 2024-11-29
> **A friendly reminder**
>
> We are grateful for the reviewer's valuable comments. Since the rest time of the discussion period is limited now, please let us know if you have other concerns. We are looking forward to hearing back from you.

---

> > ### Author Response · Authors · 2024-12-01
> > **A friendly reminder**
> >
> > We sincerely thank you for your valuable comments. We have diligently worked to address the concerns you raised and hope our responses are satisfactory. As the discussion period is nearing its end, we kindly request any additional feedback you might have. If you have any further questions or require additional clarification, please do not hesitate to let us know.

---

### Official Review · Reviewer_4YQ2 · 2024-11-05

**Soundness:** 3
**Presentation:** 4
**Contribution:** 3
**Rating:** 6
**Confidence:** 2

**Summary:**

The paper proposes a structure-oriented analysis method and a multi-agent reasoning system to enhance zero-shot reasoning capabilities in LLMs. The authors claim improvements in complex, multi-step tasks by leveraging a structure-based analysis inspired by human cognition, which identifies key syntactic and grammatical components of questions and uses multiple agents to ensure reasoning accuracy and factual reliability.

**Strengths:**

1. The idea is interesting. It applies syntactic and grammar parsing to guide LLM reasoning paths, mimicking human-like structured analysis. Additionally, the authors use probabilistic graphical models to represent the reasoning process, which provides an interpretable way to understand the model's decision-making.
2. Authors provides a detailed breakdown of each contribution, showing a clear understanding of the effectiveness of proposed methods.
3. Easy to follow the structure of the paper.

**Weaknesses:**

1. The evaluated LLMs are limited (4 LLMs), which may not be representative of the entire LLM space. The authors should consider evaluating more models (especially the open-sourced LLMs) to ensure the generalizability of their findings.
2. The evaluated datasets are also limited, which may not fully capture the diversity of reasoning tasks.
3. It could further discuss the computational efficiency of implementing SARA across large datasets, as the proposed method may be computationally expensive.
4. The paper focuses on reasoning capabilities, but only evaluates on QA tasks. It would be beneficial to evaluate on more diverse tasks to demonstrate the generalizability of the proposed method.
5. I recommend expanding the Related Works section to provide a more comprehensive overview of the field. Currently, the section references only a limited number of studies, which might not fully represent the breadth of research in zero-shot reasoning and multi-agent systems in LLMs (you can consider add some works about 'deciphering GPT-4-o1').
6. The analysis lacks depth in discussing the limitations and reasons behind the performance of the proposed method.

I'd like to increase the score if the authors address my concerns.

**Questions:**

1. The probabilistic model on Assumption 3.1 assumes independence among hidden variables in exploring reasoning paths. This assumption could be restrictive, as real-world tasks may involve dependencies between steps. How does the model handle such dependencies? Please justify the applicability in complex reasoning tasks.
2. The structure-oriented analysis relies on syntactic and grammatical patterns, what happens when the questions are ambiguous or lack clear structures? How does the model adapt to such scenarios?
3. What if the Retrieval Agent provides information that conflicts with the Reason Agent’s initial understanding? Could you elaborate more?

---

> ### Author Response · Authors · 2024-11-21
> **Formal response 1/**
>
> We thank the reviewer for the valuable comments and suggestions. We would like to address them as follows:
> 1. **[W1] Additional experiments on more models**
>
> We include two open-source models: Mixtral-8*7B and GLM-4-9B-chat, and test them on three datasets from three types of reasoning tasks. Results are shown in the following table.
>
> |             | Tasks      | Methods |             |             |               |             |                  |       |
> |-------------|------------|---------|-------------|-------------|---------------|-------------|------------------|-------|
> |             |            | Vanilla | ICL(6-shot) | CoT(6-shot) | ReAct(6-shot) | CoK(6-shot) | CoT-SC@10(0-shot ) | SARA  |
> | Mixtral-8*7 | HotpotQA   | 35.8%   | 36.1%       | 43.5%       | 53.7%         | 51.2%       | 40.4%            | 58.1% |
> |             | GSM8K      | 54.5%   | 60.2%       | 74.5%       | 79.2%         | 75.1%       | 65.9%            | 81.7% |
> |             | StrategyQA | 55.8%   | 62.9%       | 70.6%       | 77.9%         | 76.4%       | 68.3%            | 79.5% |
> | GLM-4-9B    | HotpotQA   | 45.7%   | 50.2%       | 55.3%       | 62.8%         | 60.1%       | 53.5%            | 64.9% |
> |             | GSM8K      | 72.1%   | 79.8%       | 86.9%       | 89.2%         | 85.4%       | 82.7%            | 90.5% |
> |             | StrategyQA | 60.7%   | 63.5%       | 74.3%       | 76.7%         | 78.5%       | 70.1%            | 80.3% |
>
> According to these results, our proposed method consistently outperforms baselines, suggesting a good generalization among models. We also include these results in the revision Appendix F.2.
>
> 2. **[W2,3,4] Additional experiments on diverse tasks and datasets, discussion about computation cost**
>
> Thanks for your comment. We have updated some additional experiments for different tasks and datasets as well as the computation cost as follows (same tables as the global response).
>
> | | Dataset|Vanilla| ICL(6-shot)|CoT(6-shot)|ReAct(6-shot)|CoK(6-shot)|CoT-SC@10(0-shot)|SARA|
> |-|-|-|-|-|-|-|-|-|
> |GPT4|GSM8K|66.8%|66.9%|92.1%|93.7%|91.9%|87.8%|94.2%|
> | |MATH|43.1%|55.4%|69.2%|67.5%|68.6%|64.1%|68.2%|
> | |StrategyQA|65.6%|68.1%|82.9%| 81.7%| 83.2%| 81.4%| 86.4%|
> |Qwen-max|GSM8K|68.6%|72.8%|87.5%|89.2%|87.6%|84.2%|91.3%|
> | |MATH|42.8%|45.6%|64.9%|64.5%|65.3%|61.9%|64.7%|
> | |StrategyQA|73.4%|75.5%|89.6%|88.4%|90.5%|83.1%|90.7%|
> |Qwen2-57B|GSM8K|54.9%|59.2%|82.7%|83.9%|83.5%|74.5%|84.4%|
> | |MATH|30.1%|33.5%|46.2%|47.3%|46.8%|40.8%|46.5%|
> | |StrategyQA|58.4%|63.2%|85.1%|89.2%|88.3%|79.1%|91.5%|
> |Llama3-70B|GSM8K|55.3%|58.3%|83.7%|86.5%|87.2%|76.8%|89.7%|
> | |MATH|30.7%|32.4%|42.9%|46.3%|44.9%|36.4%|44.2%|
> | |StrategyQA|57.9%|65.1%|84.2%|85.2%|85.8%|80.5%|87.1%|
>
> | |HotpotQA| |FEVER| |
> |-|-|-|-|-|
> | |Input|Output|Input|Output|
> |ReAct|1632|451|862|338|
> |CoK|791|379|587|291|
> |CoT-SC@10(0-shot)|276|2249|85|2057|
> |SARA|462|716|476|599|
>
> In short, our method performs the best in GSM8K and StrategyQA compared to other methods and achieves a comparably good performance in MATH. We conjecture that the structure-oriented analysis by its nature focuses more on the language, thus its performance in MATH is not as superior as in the other tasks. For the computational cost, our method has fewer input tokens than few-shot methods and generates fewer tokens than 0-shot methods, and is affordable considering the overall cost.
>
> 3. **[W5] Discussion on related works**
>
> We have updated Section 2 (Related Work) in the revision to add more literature about zero-shot reasoning and multi-agent systems. To name a few, we added [1,2,3,4]. We have also discussed OpenAI-o1 [5] in the revision.
>
> [1] ​​Distilling Reasoning Capabilities into Smaller Language Models.
>
> [2] Ghost in the Minecraft: Generally Capable Agents for Open-World Environments via Large Language Models with Text-based Knowledge and Memory
>
> [3] ToRA: A Tool-Integrated Reasoning Agent for Mathematical Problem Solving
>
> [4] Language Agent Tree Search Unifies Reasoning, Acting, and Planning in Language Models
>
> [5] Evaluation of OpenAI o1: Opportunities and Challenges of AGI
>
>
> 4. **[W6] Discussion on limitation**
>
> We have updated Section 7 to discuss the limitations of the proposed method. Briefly speaking, due to the nature of the structure-oriented analysis, our current method works better on problems that are clearly described in natural languages (e.g., GSM8K) than problems with pure symbol expressions (e.g., some problems in MATH). It is an interesting direction to enable models to effectively extract structures within the symbolic expressions. Besides, some specific components such as tool-agent involving tools like code executor or calculator can also be included to overcome such limitations to further improve the symbolic reasoning performance.

---

> > ### Author Response · Authors · 2024-11-21
> > **Formal response 2/**
> >
> > 5. **[Q1] Discussion on the independence assumption**
> >
> > Thank you for asking about Assumption 3.1. We have double-checked our proof procedure and figured out that the independence assumption can be removed. Intuitively, as long as the correct variables can be reached, no matter whether the states are dependent or not, it is more likely to reach the target state. In addition, the technical steps used in the proof all follow the original definition of conditional probability and conditional mean, and the formulas still apply when the variables are dependent. We have updated the theorem accordingly in our revision.
> >
> > 6. **[Q2] Elaborate on ambiguous and unclear questions**
> >
> > Ambiguity arises when the context or intent of the question is unclear, making it even more difficult for humans to provide a precise answer. In such cases, humans and LLMs tend to make predictions based on their own interpretations, which may not always align with the expected output.
> >
> > In practice, both cases—whether the question is ambiguous or lacks proper structure—- can be easily handled by asking the user to restate or clarify their question. By doing so, we can obtain a clear question and utilize our method. From an engineering perspective, adding functionality that prompts the user to rephrase unclear queries is straightforward and could help improve the overall user experience.
> >
> > From another aspect, similar to handling ambiguous and structure-lacking questions, we evaluate the robustness of our proposed method: answering questions under two different adversarial attacks. The Badchain attack appends a backdoor trigger by the end of the query, which degrades the query's clarity and breaks the query's structure. The other preemptive attack, which appends malicious answers to the queries, deliberately increases the ambiguity of the query. SARA demonstrates strong robustness against these two attacks, as shown in the last subsection of our experiment section, showcasing the effectiveness and tolerance of SARA compared to other methods.
> >
> > 7. **[Q3] Discussion on the conflict of knowledge**
> >
> > The understanding provided by Reason Agent includes structure analysis and reasoning trajectories which consist of reasoning steps towards final solutions. In this case, the retrieved knowledge is used to help generate reasoning steps if the internal knowledge of the Reason Agent is not sufficient for reasoning. In Section 4.1, when we introduce the Reason Agent, we mention that Reason Agent is prompted to determine whether external information is needed. This means that we first consider the internal knowledge of the agent, and if the internal knowledge is insufficient, external knowledge will be retrieved. From this design, we indeed avoid the potential conflict between internal and external knowledge. We have revised our manuscript and added a small note when introducing the Retrieval Agent in Section 4.1 to clarify this.

---

> ### Comment · Reviewer_4YQ2 · 2024-11-26
>
> Thank you to the authors for thoroughly addressing my concerns. Based on your detailed responses, I am raising my score to 6.

---

> > ### Author Response · Authors · 2024-11-27
> >
> > We thank the reviewer for the support and appreciation of our work!

---

### Author Response · Authors · 2024-11-21
**Global Response (1/)**

We sincerely appreciate the reviewers in reviewing our paper and providing constructive suggestions. In addition to our detailed response specifically to each reviewer, we summarize the updates of new experiments and the key updates in the revision below. All updates in the paper are colored in blue.

There are new experiment results:
* A common concern raised by all reviewers is the performance of our method in other datasets, especially those related to math reasoning. Therefore, we include two additional types of tasks and corresponding datasets: math reasoning task (GSM8K and MATH) and commonsense reasoning (stretagyQA), as shown in the following table. We also update in Table 1,2 in the revision.

| | Dataset|Vanilla| ICL(6-shot)|CoT(6-shot)|ReAct(6-shot)|CoK(6-shot)|CoT-SC@10(0-shot)|SARA|
|-|-|-|-|-|-|-|-|-|
|GPT4|GSM8K|66.8%|66.9%|92.1%|93.7%|91.9%|87.8%|94.2%|
| |MATH|43.1%|55.4%|69.2%|67.5%|68.6%|64.1%|68.2%|
| |StrategyQA|65.6%|68.1%|82.9%| 81.7%| 83.2%| 81.4%| 86.4%|
|Qwen-max|GSM8K|68.6%|72.8%|87.5%|89.2%|87.6%|84.2%|91.3%|
| |MATH|42.8%|45.6%|64.9%|64.5%|65.3%|61.9%|64.7%|
| |StrategyQA|73.4%|75.5%|89.6%|88.4%|90.5%|83.1%|90.7%|
|Qwen2-57B|GSM8K|54.9%|59.2%|82.7%|83.9%|83.5%|74.5%|84.4%|
| |MATH|30.1%|33.5%|46.2%|47.3%|46.8%|40.8%|46.5%|
| |StrategyQA|58.4%|63.2%|85.1%|89.2%|88.3%|79.1%|91.5%|
|Llama3-70B|GSM8K|55.3%|58.3%|83.7%|86.5%|87.2%|76.8%|89.7%|
| |MATH|30.7%|32.4%|42.9%|46.3%|44.9%|36.4%|44.2%|
| |StrategyQA|57.9%|65.1%|84.2%|85.2%|85.8%|80.5%|87.1%|

According to the results above, our method (SARA) shows good generalization ability on various reasoning tasks and datasets. Our method outperforms baselines in both GSM8K and StretagyQA datasets, suggesting that SARA can handle math and commonsense reasoning tasks.

In terms of the MATH dataset, the performance of SARA is slightly worse than the best method. We conjecture that SARA, as well as the structure-oriented analysis behind it, focuses more on the language in the data, thus its performance in MATH is not as superior as in the other tasks. It is an interesting future direction to study how to better handle symbolic representations in math problems.
* The new column in the above table (also Table 1, 2 in the revised paper) is the results of Zero-shot CoT-SC@10, an additional method suggested by Reviewer i9ee. Specifically, we repeat Zero-shot CoT 10 times and do the majority voting (helped by an LLM) to select the answer. As the table above shows, SARA has consistently higher accuracy than Zero-shot CoT-SC@10. This is mainly due to the help of the structure-oriented analysis, which can provide guidance for reasoning. In contrast, Zero-shot CoT-SC@10 relies on inherent randomness.

* Another comment from **Reviewer i9ee**, **4YQ2** and **N64Z** is about the computation overhead. The major computation overhead in SARA is the structure-oriented analysis and its refinement. We include a computation cost analysis on HotpotQA and FEVER for illustration as in the following table. We report both the average number of input and output tokens.

| |HotpotQA| |FEVER| |
|-|-|-|-|-|
| |Input|Output|Input|Output|
|ReAct|1632|451|862|338|
|CoK|791|379|587|291|
|CoT-SC@10(0-shot)|276|2249|85|2057|
|SARA|462|716|476|599|

According to this table, our method has fewer input tokens than the few-shot method since we do not need few-shot examples. Additionally, our method also generates fewer tokens than Zero-shot CoT SC@10. Together with the fact that the price for GPT-4 is \\$0.03 for 1k input token and \\$0.06 for 1k output token, SARA is affordable compared with baselines and achieves a good balance between tokens and effectiveness considering its superior performance in most tasks. We have included this discussion in the revision in Appendix G.
* Open-source models (**Reviewer 4YQ2**): We include two open-source models: Mixtral-8*7B and GLM-4-9B-chat, and test them on three datasets from three types of reasoning tasks. Results are shown in the following table. From this table, our method consistently outperforms better than baselines, showing the generalization over other models.

| |Datasets|Vanilla|ICL(6-shot)|CoT(6-shot)|ReAct(6-shot)|CoK(6-shot)|CoT-SC@10(0-shot)|SARA|
|-|-|-|-|-|-|-|-|-|
|Mixtral-8*7B|HotpotQA|35.8%|36.1%|43.5%|53.7%|51.2%|40.4%| 58.1%|
| |GSM8K|54.5%|60.2%|74.5%|79.2%|75.1%|65.9%|81.7%|
| |StrategyQA|55.8%|62.9%|70.6%|77.9%|76.4%|68.3%|79.5%|
|GLM-4-9B|HotpotQA|45.7%|50.2%|55.3%|62.8%|60.1%|53.5%|64.9%|
| |GSM8K|72.1%|79.8%|86.9%|89.2%|85.4%|82.7%|90.5%|
| |StrategyQA|60.7%|63.5%|74.3%|76.7%|78.5%|70.1%|80.3%|

---

> ### Author Response · Authors · 2024-11-21
> **Global Response (2/)**
>
> Below is a summary of the major changes in our revision:
>
> - Experiments (**Reviewer 4YQ2, 57Cm, i9ee, 8je1, N64Z, uEkW**):
>     - The news results on GSM8K, MATH, and strategyQA have been updated in Table 1 and 2.
>     - We added the Zero-shot CoT-SC@10 as an additional baseline in our experiments.
>     - The computation overhead has been updated in Appendix G.
>     - New results in additional open-source models (Mixtral and GLM), in Table 7, Appendix F.2.
> - Limitation (**Reviewer 4YQ2, 57Cm, 8je1, N64Z**): We have updated Section 7 to discuss the limitations of the proposed method. In summary, due to the nature of the structure-oriented analysis, SARA works better on problems that are clearly described in natural languages (e.g., GSM8K) than problems with symbolic expressions (e.g., some problems in MATH). An interesting future direction is to explore ways to enable models to effectively extract structures within the symbolic expressions. Besides, some specific components such as tool-agent involving tools like code executor or calculator can also be included to overcome such limitations to further improve the symbolic reasoning performance.
> - Theoretical results in Section 3.2: Since we have included additional experimental results in our revision and Reviewer uEkW also commented that our theoretical insights in Section 3.2 can be improved in terms of presentation, we have re-arranged Section 3.2 and moved its detailed and formal theoretical analysis to Appendix A. In the revision, Section 3.2 is only 1 page and provides the basic intuition and an informal theoretical statement. We hope this revision improves the clarity and logical flow of this paper.
> - Additional related works (**Reviewer 4YQ2, i9ee**): We updated Section 2 (Related Work) to incorporate the suggestions of the reviewers.  A more detailed related work comparison can be found in the response to **Reviewer 4YQ2** and **i9ee**.
>
> - We also added clarifications in the revision to clarify some comments of reviewers, e.g., the novelty of this paper (**Reviewer 57Cm**) in Section 1 and knowledge conflict of the Retrieval Agent (**Reviewer 4YQ2**) in Section 4.2.

---

### Meta-Review · Area_Chair_oVBy · 2024-12-24

**Metareview:**

Summary: The paper introduces a method to improve zero-shot reasoning in Large Language Models (LLMs) by using a structure-oriented analysis inspired by human reasoning. This approach focuses on understanding the syntactic and grammatical structures of problem statements to enhance reasoning processes. Additionally, the authors propose the Structure-Oriented Autonomous Reasoning Agents (SARA) framework, a multi-agent system that incorporates reasoning, refinement, retrieval of external knowledge, and shared memory. The paper provides both theoretical and empirical support, showing that SARA improves performance on complex reasoning tasks, surpasses some few-shot baselines, and demonstrates robustness against adversarial attacks.

Strengths:
- Intuitive method to break down complex tasks (but not very novel)
- PGM based analysis
- Robustness to attacks
- The paper is well-organized and easy to follow

Weakness:
- Lukewarm response from all but one reviewer and the positive reviewer didn't champion the paper
- Limited novelty: Many reviewers felt the design had similarities to existing work
- Lack of comprehensive evaluation: Initially limited to only QA tasks, not providing clear compute-cost trade-offs, etc.
- Limited performance on mathematical reasoning tasks (MATH dataset) compared to few-shot methods
- Heavy reliance on LLM's inherent capabilities for structure analysis
- Over claiming: the title and abstract are very generic but the work does not cover all forms of reasoning


Decision: Given the lack of enthusiasm from the reviewers and limited novelty, and limited improvements in performance, unfortunately, the paper can't be accepted in its current form and addressing all the concerns would warrant another round of reviewing.

**Additional Comments On Reviewer Discussion:**

We thank the authors and reviewers for engaging during the discussion phase towards improving the paper. Below are some of the highlights:

1. Limited evaluation scope:
- A common concern raised by all reviewers is the performance of our method in on more datasets and also open-sourced models
- Added extensive experiments on GSM8K, MATH, StrategyQA
- Included results with additional open-source models

2. Novelty and contribution:
- Better articulated differences from existing work
- Emphasized focus on zero-shot reasoning improvement
- Clarified relationship between structure analysis and agent design
- Still reviewers were not totally satisfied

3. Computation costs:
- Added detailed analysis of input/output tokens
- Showed competitive efficiency versus baselines

4. Theoretical framework concerns:
- Restructured Section 3.2 to better integrate with main contributions
- Moved detailed proofs to appendix
- Clarified relationship to structure-oriented analysis

Despite authors best effort, even the reviewers who were responsive were convinced.

---

### Decision · Program_Chairs · 2025-01-22

Reject